# Cryo-EM structures of the Spo11 core complex bound to DNA

You Yu[1,2,9], Juncheng Wang[1,3,9], Kaixian Liu ⓘ [4,9], Zhi Zheng[4,5,9], Meret Arter[4], Corentin Claeys Bouuaert ⓘ [4,6,7], Stephen Pu[4,6,8], Dinshaw J. Patel ⓘ [1,5] ✉ & Scott Keeney ⓘ [4,5,6] ✉

DNA double-strand breaks that initiate meiotic recombination are formed by the topoisomerase-relative enzyme Spo11, supported by conserved auxiliary factors. Because high-resolution structural data have not been available, many questions remain about the architecture of Spo11 and its partners and how they engage with DNA. We report cryo-electron microscopy structures at up to 3.3-Å resolution of DNA-bound core complexes of *Saccharomyces cerevisiae* Spo11 with Rec102, Rec104 and Ski8. In these structures, monomeric core complexes make extensive contacts with the DNA backbone and with the recessed 3′-OH and first 5′ overhanging nucleotide, establishing the molecular determinants of DNA end-binding specificity and providing insight into DNA cleavage preferences in vivo. The structures of individual subunits and their interfaces, supported by functional data in yeast, provide insight into the role of metal ions in DNA binding and uncover unexpected structural variation in homologs of the Top6BL component of the core complex.

Spo11 is related to the DNA-cleaving A subunit (Top6A) of archaeal topoisomerase VI (Topo VI)[1]. Both enzymes cut DNA through a trans-esterification reaction in which a tyrosine side chain severs the DNA backbone and attaches covalently to the 5′ terminus by a tyrosyl phosphodiester linkage. Two Top6A or Spo11 proteins work together to make a double-strand break (DSB) with two-nucleotide 5′ overhangs[2,3].

Topo VI is a heterotetramer of two A and two B subunits. Top6A has a winged-helix (WH) domain that carries the DNA-cleaving tyrosine and a metal-ion-binding Rossmann fold known as the Toprim domain[4–6] (Fig. 1a). Scission of each DNA strand involves the tyrosine of one Top6A monomer interacting with the $Mg^{2+}$-binding pocket of the second Top6A monomer to form a hybrid active site, thus requiring dimer formation. Sequence alignments and computational modeling predict equivalent domains in Spo11 (refs. 7–10) (Fig. 1a,b).

Top6B has an N-terminal GHKL-family adenosine triphosphate (ATP) hydrolase (ATPase) domain that dimerizes upon binding to ATP and a 'transducer' domain consisting of a β-sheet backed by α-helices, one of which provides a long lever arm connecting the GHKL domain to the Top6A interface (Fig. 1a)[4,5,11]. Mouse and plant Top6B homologs (Top6BL) have the transducer and GHKL folds but the homologs from some species, including yeasts, resemble only the transducer[1,12,13]. Top6B also has a helix-2–turn–helix (H2TH) domain and sometimes a C-terminal domain (CTD), of unknown function. The eukaryotic homologs appear to lack these domains.

In *Saccharomyces cerevisiae*, Spo11 forms a 'core complex' with Rec102, Rec104 and Ski8 (ref. 14). Core complexes are found in most eukaryotic taxa but with substantial diversity[15–18]. Rec102 is homologous to the Top6B transducer but lacks a clear GHKL equivalent[1,16]

[1]Structural Biology Program, Memorial Sloan Kettering Cancer Center, New York, NY, USA. [2]Centre for Infection Immunity and Cancer (IIC), Zhejiang University–University of Edinburgh Institute, Zhejiang University School of Medicine, Haining, China. [3]Advanced Medical Research Institute, Cheeloo College of Medicine, Shandong University, Jinan, China. [4]Molecular Biology Program, Memorial Sloan Kettering Cancer Center, New York, NY, USA. [5]Louis V. Gerstner Graduate School of Biomedical Sciences, Memorial Sloan Kettering Cancer Center, New York, NY, USA. [6]Howard Hughes Medical Institute, Memorial Sloan Kettering Cancer Center, New York, NY, USA. [7]Louvain Institute of Biomolecular Science and Technology, Université catholique de Louvain, Louvain-La-Neuve, Belgium. [8]Present address: WaypointBio, New York, NY, USA. [9]These authors contributed equally: You Yu, Juncheng Wang, Kaixian Liu, Zhi Zheng. ✉e-mail: pateld@mskcc.org; s-keeney@ski.mskcc.org

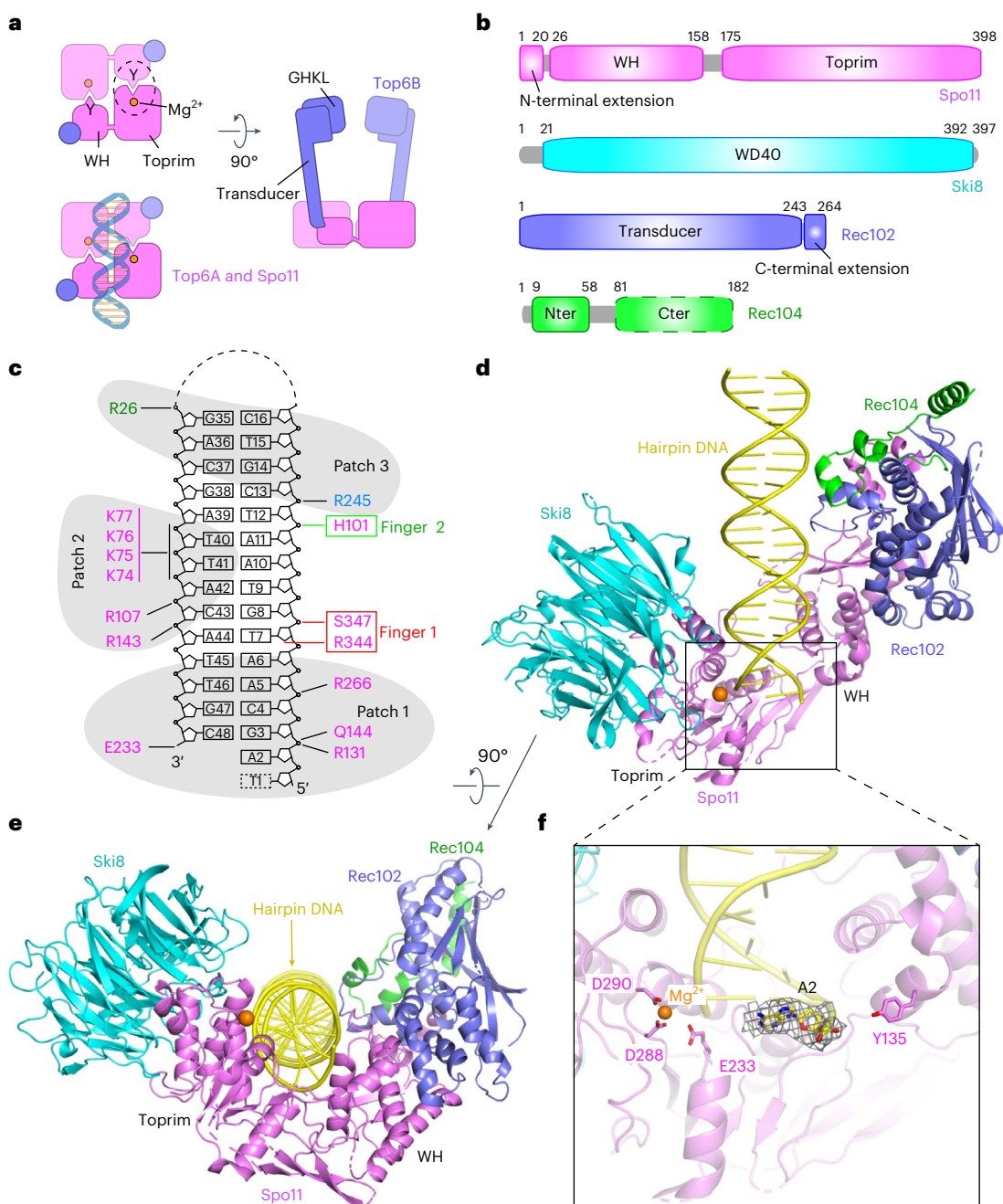

**Fig. 1 | Cryo-EM structure of Spo11 core complex bound to hairpin DNA.**
**a**, Schematic of tertiary and quaternary organization of the Topo VI holoenzyme. Left, top views (with and without DNA) looking down into the DNA-binding channel of Top6A (or Spo11), illustrating the dimer interface, catalytic tyrosine (Y), metal-binding pocket and hybrid active site (dashed circle). Right, side view. **b**, Domain organization of Spo11, Ski8, Rec102 and Rec104. **c**, Hairpin-DNA sequence and intermolecular contacts between the DNA (mostly

sugar-phosphate backbone) and amino acid side chains of Spo11 (magenta), Rec102 (blue) and Rec104 (green). Groups of DNA-interacting residues described in the text (patches and fingers) are highlighted. **d**, Ribbon representation of the 3.7-Å-resolution cryo-EM structure. **e**, As in **d** but rotated 90°. **f**, Detailed view of Spo11 and the 5′ overhang end of the hairpin DNA, highlighting select active site residues. The density of the first nucleotide of the overhang (A2) is shown as gray mesh. Cter, C-terminal; Nter, N-terminal.

(Fig. 1b). GHKL loss has occurred in multiple eukaryotic lineages[16]. Moreover, Rec104 is enigmatic; its narrow phylogenetic distribution[15] and consequent shallow multiple-sequence alignment preclude a high-confidence prediction by AlphaFold2. It has been hypothesized that Rec104 may have either evolved from or replaced the GHKL domain[1,13,14,16].

Purified recombinant core complexes have a 1:1:1:1 stoichiometry, which was unexpected because Top6A forms a stable dimer[4–6] and because DNA cleavage requires hybrid active sites. The monomeric core

complexes bind noncovalently with high affinity (subnanomolar dissociation constant ($K_d$)) to DNA ends that mimic the cleavage product in having a two-nucleotide 5′ overhang[14]. This tight binding may allow Spo11 to cap DSB ends, potentially affecting subsequent repair[14,19,20].

Crystal structures have been reported for Topo VI (refs. 4–6,11) but not for eukaryotic Spo11. Homology models for yeast core complexes templated on Topo VI (refs. 12–14) or predicted by RosettaFold and AlphaFold2 (ref. 9) differ from one another in terms of subunit tertiary structure and protein–protein contacts[9]. Moreover, no structure

determination has been reported for any Spo11 or Topo VI relative bound to DNA. We address these issues with cryo-electron microscopy (cryo-EM) structures of DNA-bound yeast Spo11 core complexes.

## Results and discussion

### Cryo-EM structure of the core complex bound to hairpin DNA

*S. cerevisiae* core complexes were conformationally variable in negative-stain EM but adopted a more uniform structure when bound to DNA[14]. We leveraged the tight binding to DNA ends to determine a cryo-EM structure of core complexes bound to a 23-bp hairpin DNA with a two-nucleotide 5′ overhang. Spo11 binds tightly to the free end of this substrate and with much lower affinity to the hairpin end[14].

Core complexes were purified after expression in insect cells (Extended Data Fig. 1a) and incubated with Mg[2+] and hairpin DNA (Fig. 1c and Extended Data Fig. 1b). The structure of the protein–DNA complexes was solved at 3.7-Å resolution. The cryo-EM reconstruction is shown in Extended Data Fig. 2a–d, cryo-EM statistics are provided in Table 1 and the density representation is presented in Extended Data Fig. 2e. The model was built using subunit structures predicted by AlphaFold2 (ref. 10) and manually refined by positioning of bulky amino acid side chains.

The core complex is monomeric (1:1:1:1 stoichiometry) and adopts a V shape with Spo11 at the base contacting the three other subunits (Fig. 1d,e). The hairpin's overhang end is anchored in a deep cleft between the Spo11 WH and Toprim domains (Fig. 1d). The remainder of the duplex makes non-sequence-specific contacts primarily with Spo11 and fewer with Rec102 and Rec104 but not with Ski8 (Fig. 1e). The proteins form a left-handed wrap around the DNA. Protein–DNA contacts in the complex are summarized in Fig. 1c.

We could trace the base, sugar and 5′-phosphate of the first nucleotide (A2) in the overhang adjacent to the terminal base pair. By contrast, the next overhang base (T1) could not be traced because of weak density. Residue Y135 involved in DNA covalent bond formation is on the opposite side of the DNA end from the Mg[2+]-coordinating E233, D288 and D290 residues (Fig. 1f). This separation of active site components is expected given that, with only one copy of Spo11, the structure shows the configuration of two catalytic half-sites (Fig. 1a). It is likely that the observed structure mimics the post-DSB product complex, except that Y135 would be covalently bound to the 5′ end in a true product complex.

### Cryo-EM structure of the core complex bound to gapped DNA

We also solved the structure of the core complex bound to a double-hairpin substrate with a single-stranded DNA (ssDNA) gap (Fig. 2a and Extended Data Fig. 1c) at 3.3-Å resolution. The cryo-EM reconstruction is shown in Extended Data Fig. 3, cryo-EM statistics are provided in Table 1 and the density maps are presented in Extended Data Fig. 4a–e. Two views are shown in Fig. 2b,c. The preference for binding to bent DNA[14] motivated these experiments, on the premise that the ssDNA region would provide the flexibility needed to assemble dimeric complexes that resemble the pre-DSB state. However, the majority of particles again had just a single core complex (Extended Data Fig. 3a).

We can trace the duplex of one hairpin plus the first nucleotide of the ssDNA (A39) including its 5′-phosphate. The bound hairpin is the one with a free 3′-OH and ssDNA extending from the 5′ end (Fig. 2a), which is structurally analogous to the single hairpin initially used. Weak density in the unsharpened map spanned the single-stranded nucleotides T38-T37-A36 and a portion of the second hairpin, with the two duplex segments aligned at a relative angle of 130° (Fig. 2d). Protein–DNA contacts are summarized in Fig. 2a.

The α-helical and β-strand segments for the four subunits are shown in Extended Data Fig. 5. Spo11 is a central hub making extensive protein–protein and protein–DNA contacts. The two arms of the V-shaped protein scaffold comprise the Spo11 WH domain interacting with Rec102 and Rec104 and the Toprim domain interacting with Ski8 (Fig. 2b,c). The structures with hairpin DNA and gapped DNA superpose

## Table 1 | Cryo-EM data collection, refinement and validation statistics

| | Spo11 core complex bound to hairpin DNA (EMD-42501), (PDB 8URU) | Spo11 core complex bound to gapped DNA (EMD-42497), (PDB 8URQ) |
|---|---|---|
| **Data collection and processing** | | |
| Magnification | 22,500 | 22,500 |
| Voltage (kV) | 300 | 300 |
| Electron exposure (e⁻ per Å²) | 53.00 | 53.00 |
| Defocus range (µm) | −1.0 to −2.5 | −1.0 to −2.5 |
| Pixel size (Å) | 1.064 | 1.064 |
| Symmetry imposed | *C1* | *C1* |
| Initial particle images (no.) | 1,829,931 | 7,804,373 |
| Final particle images (no.) | 128,787 | 548,674 |
| Map resolution (Å) | 3.66 | 3.29 |
| FSC threshold | 0.143 | 0.143 |
| Map resolution range (Å) | 7.3–2.3 | 6.2–2.3 |
| **Refinement** | | |
| Initial model used | AF-Q02793-F1 AF-P23179-F1 AF-Q02721-F1 ColabFold model of Rec102+Rec104 | Spo11 core complex bound to hairpin DNA |
| Model resolution (Å) | 3.6 | 3.3 |
| FSC threshold | 0.5 | 0.5 |
| Model resolution range (Å) | 3.93–3.51 | 3.63–3.19 |
| Map sharpening *B* factor (Å²) | −158.0 | −155.8 |
| Model composition | | |
| Nonhydrogen atoms | 8,895 | 8,569 |
| Protein residues | 1,001 | 993 |
| Nucleotides | 43 | 31 |
| *B* factors (Å²) | | |
| Protein (mean) | 76.12 | 47.85 |
| Ligand (mean) | 144.09 | 20.00 |
| R.m.s.d. | | |
| Bond lengths (Å) | 0.005 | 0.017 |
| Bond angles (°) | 0.979 | 1.250 |
| **Validation** | | |
| MolProbity score | 2.80 | 2.84 |
| Clashscore | 20 | 18 |
| Poor rotamers (%) | 3.21 | 4.61 |
| Ramachandran plot | | |
| Favored (%) | 88.11 | 88.88 |
| Allowed (%) | 10.95 | 10.81 |
| Disallowed (%) | 0.94 | 0.31 |

well (root-mean-square deviation (r.m.s.d.) = 1.23 Å); hence, the following sections focus on the higher-resolution gapped-DNA structure.

### Validation of the Spo11–Ski8 interface

The Spo11–Ski8 interface was previously modeled using a crystal structure of the Ski complex, in which two Ski8 copies contact two copies of a QRxxΦ motif in Ski3 that is also present in fungal Spo11 proteins[14,21]. This motif is critical for Spo11–Ski8 interaction in *S. cerevisiae* both

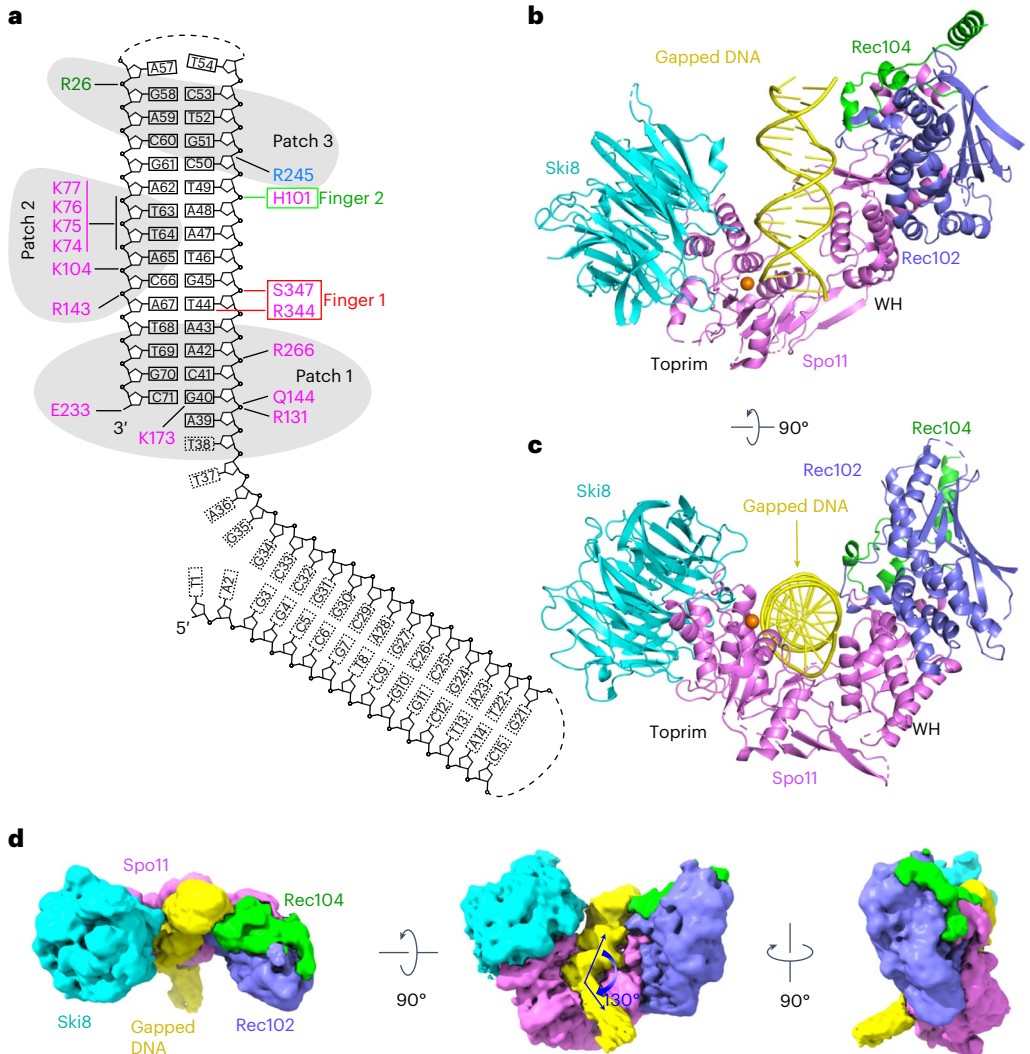

**Fig. 2 | Cryo-EM structure of Spo11 core complex bound to gapped DNA. a**, Gapped-DNA sequence and intermolecular contacts between the DNA and amino acid side chains of Spo11 (magenta), Rec102 (blue) and Rec104 (green). **b**, Ribbon representation of the 3.3-Å-resolution cryo-EM structure. **c**, As in **b** but rotated 90°. **d**, Three views of the unsharpened cryo-EM map.

in vivo and in vitro[14,22]. This same interface was also predicted computationally[9]. The cryo-EM structure matched these predictions, showing that Ski8 recognizes $Q_{376}$REIFF in the Spo11 Toprim domain (Extended Data Fig. 6a). The interface involves both hydrophobic and hydrogen-bonding interactions (Extended Data Fig. 6b,c), including a computationally predicted extended interaction surface (Extended Data Fig. 6d).

**Rec102 and its interaction with Spo11**

The overall core complex architecture resembles Top6A–Top6B, as expected (Fig. 3a). Rec102 comprises a six-stranded β-sheet extended by multiple α-helices (Figs. 2b and 3b and Extended Data Fig. 5c). It has structural elements in common with the Top6B transducer domain, as anticipated, but with substantial differences. We consider here five main features from Top6B: a central β-sheet, a WKxY motif, a 'switch loop' that interacts with the ATP-binding site in the GHKL domain, the α-helical lever arm ('stalk') and the interface with Top6A (Fig. 3b, top). Conserved β-strands are indicated by uppercase letters ordered by tertiary-structure position, while numerical designations for β-strands correspond to primary-structure position (for example, Extended Data Fig. 5c); thus, these differ between species. Our structure agrees with crosslinking data[14] where K60 and K64 in β-strand D crosslinked with K79 in nearby α-helix 1.

In Top6B, the central β-sheet is a scaffold supporting the stalk helix and the GHKL and H2TH domains[4,5,11]. Rec102 has a similar β-sheet, with the first four strands corresponding to segments of homology with Top6B (ref. 13) (Fig. 3b, strands labeled A–D, and Extended Data Fig. 7a,b). These strands are mostly encoded in *REC102* exon 1, which is essential in vivo[23]. The sheet is extended by one (Top6B) or two (Rec102) additional β-strands but these arise differently in the sequence; the short strand in Top6B is immediately after strand D but the two longer strands in Rec102 instead occur after a conserved helix (Fig. 3b).

Another conserved element is a tryptophan within a motif that has the sequence WKxY in archaea[13] and that starts at the end of a set of short α-helices and continues into the subsequent turn (Fig. 3b and Extended Data Fig. 7b). The lysine in this motif is part of a putative DNA-binding surface in Topo VI that is critical for activity[24]. The tryptophan is in the hydrophobic core of *Saccharolobus* Top6B, packing against the stalk and hydrophobic residues at the beginning of strand C (Extended Data Fig. 7c). The tryptophan is nearly invariant in eukaryotes but the rest of the motif is highly variable ($W_{91}$EEQ in Rec102) (Extended Data Fig. 7b). In Rec102, W91 packs against the equivalent of the stalk helix, analogous to Top6B, but contacts different strands in the β-sheet: I59 in strand D (β4) and L113 in strand E (β6) (Extended

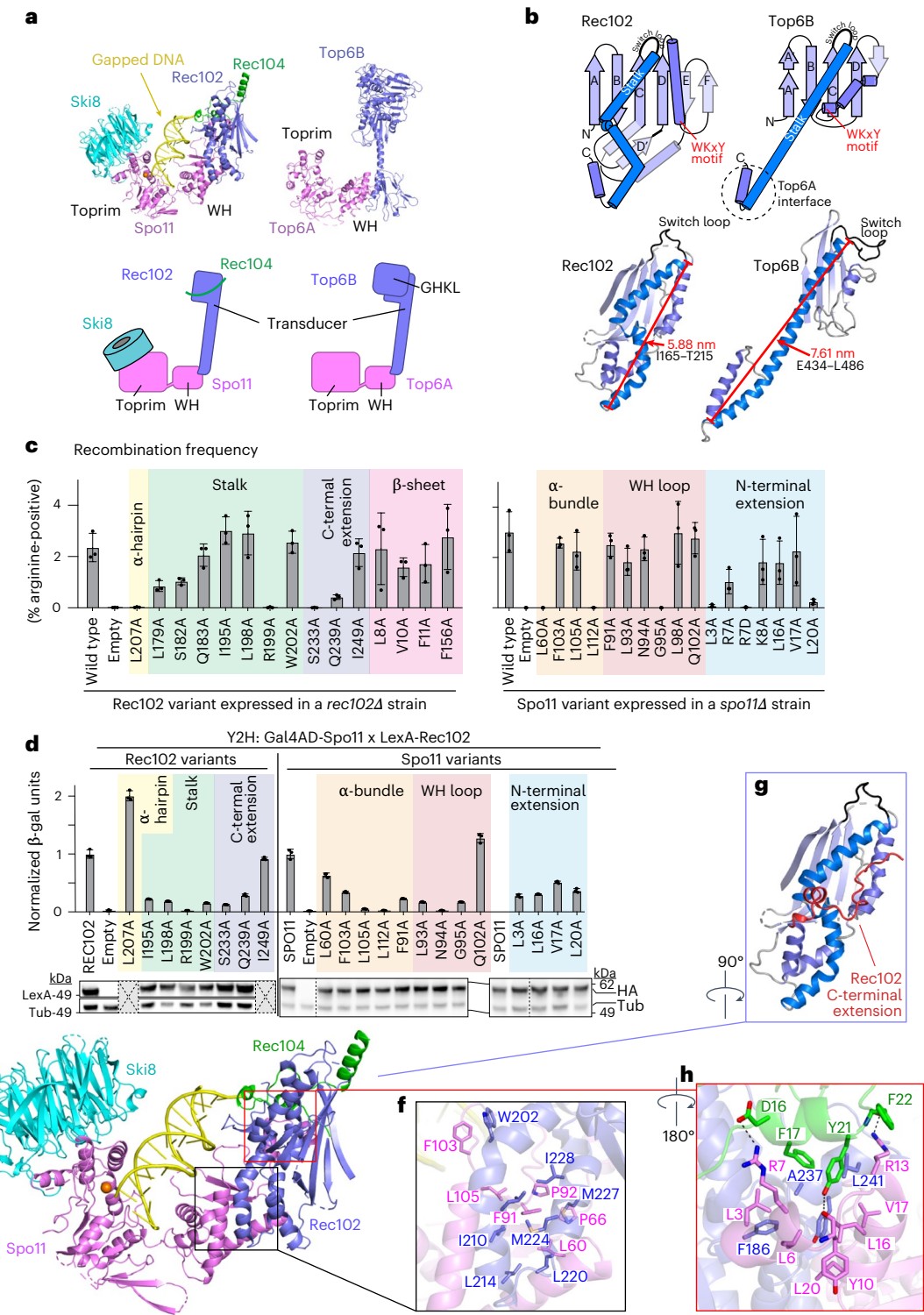

**Fig. 3 | Rec102 folding and interactions with other subunits. a**, Comparison of overall architectures of the Spo11 core complex and Topo VI. Top, ribbon representations of the cryo-EM structure from this study (left) and the Top6a–Top6b heterodimer from a crystal structure of the *Methanosarcina mazei* Topo VI holoenzyme (right; PDB 2Q2E)[5]. Bottom, schematic of tertiary and quaternary organization. **b**, Comparison of transducer domains of Rec102 and Top6B. Top, protein topology schematics of Rec102 and Top6B highlighting conserved structural elements. Bottom, ribbon representations of Rec102 (this study) and Top6B (PDB 2ZBK)[4], with the same color settings as in the topology schematics. **c**, Heteroallele recombination assays with different Rec102 (left) or Spo11 (right) variants introduced into *rec102Δ* or *spo11Δ* reporter strains, respectively. Bars show the mean ± s.d. of three biological replicates; points show individual measurements. Different shading colors highlight the indicated protein elements and are matched

with **d** and Extended Data Fig. 8c,e to facilitate comparisons. **d**, Quantitative β-galactosidase assays to measure Y2H interactions of mutants of Gal4AD-Spo11 and LexA-Rec102. Bars show the mean ± s.d. of three biological replicates; points show individual measurements. Immunoblots below the graph show levels of the fusion proteins in whole-cell extracts detected with either anti-HA (to detect the Gal4AD fusion) or LexA antibodies. Vertical dashed lines indicate where blot images were spliced. Gray boxes indicate select mutants that were not tested by immunoblot because they were proficient in the Y2H and/or recombination assays. **e**–**h**, Details of protein–protein contacts within the gapped-DNA-bound core complex: an overall view of the structure (**e**); zoomed details showing hydrophobic interactions between Rec102 and Spo11 (**f**); the position of the C-terminal extension of Rec102 (red; **g**); intermolecular hydrogen bonds and hydrophobic interactions between the N-terminal extension of Spo11 and Rec102 and Rec104 (**h**).

Data Fig. 7c). This motif is ≥16 Å away from the DNA; thus, it does not appear to contribute to DNA binding in the end-bound complex.

The switch loop is a 16-residue sequence in *Saccharolobus* Top6B that connects strand C with the stalk helix (Fig. 3b and Extended Data Fig. 7b). It contains a lysine that contacts ATP bound by the GHKL domain's active site and that is essential for ATPase activity in Topo VI and other topoisomerases[25–27]. Our structure confirms the earlier deduction of the position in the Rec102 sequence equivalent to the switch loop[13] but the loop is smaller in Rec102 than previously appreciated (just eight residues, $S_{157}$KEGNYVE). We tested the importance of this element by assessing the ability of *rec102* mutant alleles to complement a *rec102*-null strain in a heteroallele recombination assay, in which arginine-positive prototrophs can be generated by recombination between two different *arg4* mutant alleles (Extended Data Fig. 7d). Where tested, alanine substitutions across the switch loop had no effect on Rec102 activity in vivo and completely replacing the loop sequence with a flexible glycine or serine linker only reduced recombination about fourfold (Extended Data Fig. 7e). These results suggest that the specific amino acid sequence of the switch loop is not essential for Rec102 function, in keeping with the absence of a GHKL domain.

The stalk of Rec102 begins at residue I165, after the switch loop. As in Top6B, the first part of the stalk is an amphipathic helix whose hydrophobic face runs diagonally across strands D through A of the central β-sheet (Fig. 3b and Extended Data Fig. 7f). However, unlike the long, continuous α-helix that makes up the stalk of Top6B, the equivalent region of Rec102 is distorted into four helical segments to make a kinked path that is more compact and engages in more extensive intramolecular and intermolecular contacts within the core complex (Fig. 3b)[9]. We examined substitutions of four stalk residues predicted to contact Spo11 (I195A, L198A, R199A and W202A). Of these, R199A completely eliminated both meiotic recombination in vivo and yeast two-hybrid (Y2H) interaction with Spo11, without diminishing protein levels (green shading in Fig. 3c,d). R199 forms hydrogen bonds with multiple residues in Spo11 (Extended Data Fig. 8a). The other three substitutions reduced the Y2H interaction but did not disrupt recombination in vivo (Fig. 3c,d). Perhaps other interactions within the DSB-forming machinery can compensate when this interface with Spo11 is only partially weakened.

The Rec102 stalk winds its way to interact with the Spo11 WH domain. A helical hairpin at the end of the stalk aligns with a pair of α helices from Spo11 (Fig. 3b,e,f). The structure of this part of the Spo11–Rec102 interface resembles the computational Spo11–Rec102 model, which was earlier noted to resemble the Top6A–Top6B interface[9]. However, Spo11 and Rec102 have a much more extensive interface than Top6A and Top6B (1,681 Å² for Spo11–Rec102 versus 958 Å² for Topo VI; Extended Data Fig. 8b), mediated by both hydrophobic and hydrogen-bonding interactions (Fig. 3f and Extended Data Fig. 8a). Substitutions across the Rec102 helical hairpin (L214 and residues 220–236) were previously shown to disrupt the Y2H interaction with Spo11 and to eliminate meiotic recombination initiation in vivo[14]. Meiotic recombination was also compromised by single-alanine substitutions for interfacial residues L207 of Rec102 (yellow shading in Fig. 3c, left) and L60 and L112 in Spo11 (peach shading in Fig. 3c, right). Of these, Spo11-L112A was also deficient for the Y2H interaction with Rec102 despite having normal protein levels (Fig. 3d) but Rec102-L207A and Spo11-L60A retained the Y2H interaction, suggesting that their recombination defects are attributable to another biochemical property of the interface separate from stabilizing the interaction per se. Other tested residues in this interface were dispensable for meiotic recombination in vivo (Spo11 F103, which packs against Rec102 W202, and Spo11 L105) (Fig. 3c,d).

In addition to these elements that are conserved with Top6B, Rec102 sports a 34-residue C-terminal extension beginning at D231. This segment wraps around the stalk, contributes to the extended interface with Spo11 and also makes contacts with the DNA and Rec104

(Fig. 3e,g), as discussed below. Double-alanine substitutions for Rec102 D231;K232 or T235;T236 compromise both meiotic recombination and the Spo11 Y2H interaction[14]. Rec102 S233 and Q239 form hydrogen bonds with the Spo11 peptide backbone at G95 and L98, respectively (Extended Data Fig. 8a). Rec102-S233A, Rec102-Q239A and Spo11-G95A severely compromise meiotic recombination in vivo and diminish the Spo11–Rec102 Y2H interaction without affecting protein levels (purple shading in Fig. 3c, left, salmon shading in Fig. 3c, right and Fig. 3d).

There is also an extension (25 residues) of the N terminus of Spo11 before the WH domain, of unknown function[16]. This α-helical segment contacts Rec102's C-terminal extension and stalk and the N terminus of Rec104 through hydrophobic interactions and hydrogen bonds (Fig. 3e,h). The Spo11 extension places the N termini of Spo11, Rec102 and Rec104 in close proximity, which agrees with crosslinking data and explains why an N-terminal maltose-binding protein tag on any of these proteins resulted in extra density in nearly the same position in negative-stain EM[14]. Several substitutions in the Spo11 extension (L3A, R7D and L20A) severely compromised meiotic recombination in vivo and diminished Y2H interactions with Rec102 or Rec104 (blue shading in Fig. 3c, right and Extended Data Fig. 8c). Most of the other tested mutants in this region also decreased the Y2H interactions without affecting recombination (Fig. 3c and Extended Data Fig. 8c).

## Rec104 structure and protein interactions

Only residues 9–58 (of 182) of Rec104 are visible in the structure, forming three α-helices that interact with both Rec102 and Spo11 (Fig. 4a–c). The structure agrees with previously observed[14] crosslinks between Rec104 K43 and Rec104 K10, K36 and K39 (α-carbon-to-α-carbon distances of 12–20 Å). However, the resolution is lower for this part of the structure (Extended Data Fig. 2d and Extended Data Fig. 3d); thus, the details of interfacial contacts should be viewed cautiously.

Previous efforts to identify Rec104 orthologs failed to find them outside of *Saccharomyces* species[28]. The structure of the core complex allowed us to revisit this because the visible parts of Rec104 correspond to a previously unrecognized conserved domain (Fig. 4d). We expanded the collection of Rec104 homologs by focusing on this domain but we were still only able to detect them in family Saccharomycetaceae (Fig. 4d and Methods). Particularly well conserved residues (Fig. 4d) line the interfaces with Rec102 and Spo11, including Rec104 Y21 (Fig. 4a), Rec104 F17, L18, V23, V27, F31 and L33 (Fig. 4b) and Rec104 I41 and F46 (Fig. 4c). The C-terminal portion of Rec104 that is not visible in the cryo-EM structure is poorly conserved (Fig. 4d).

We tested the functional importance of interfacial residues. A Rec104-Y21A mutant was strongly defective for meiotic recombination (Extended Data Fig. 8d) and the Y2H interaction with Spo11 (Extended Data Fig. 8c) but was expressed at normal levels and retained the Y2H interaction with Rec102 (Extended Data Fig. 8e). There was little if any effect for the other Rec104 mutants tested (Extended Data Fig. 8d,e) or for Rec102 β-sheet substitutions L8A, V10A, F11A and F156A (pink shading in Fig. 3c, left and Extended Data Fig. 8e). Interestingly, Rec102 stalk substitutions L179A and Q183A disrupted the Y2H interaction with Rec104 but had only a modest effect (L179A) or no effect (Q183A) in recombination (green shading in Fig. 3c and Extended Data Fig. 8e). Because this Y2H assay is performed in vegetative cells, a plausible interpretation is that these substitutions weaken the Rec102–Rec104 interaction but are compensated for in the context of the full DSB apparatus.

Prior crosslinking experiments suggested that Rec104 lies near where the GHKL domain abuts the transducer domain in Topo VI, supporting the proposal that Rec104 replaces the GHKL domain[1,14]. We explored this idea using our structure.

The distances between Rec104 K43 and either Rec102 K60 or K64 agree well with crosslinking data (Extended Data Fig. 8f). Rec102 K60 and K64 are in strand D of the central β-sheet, with their side chains on the opposite face of the sheet from the stalk (Extended Data Fig. 7b

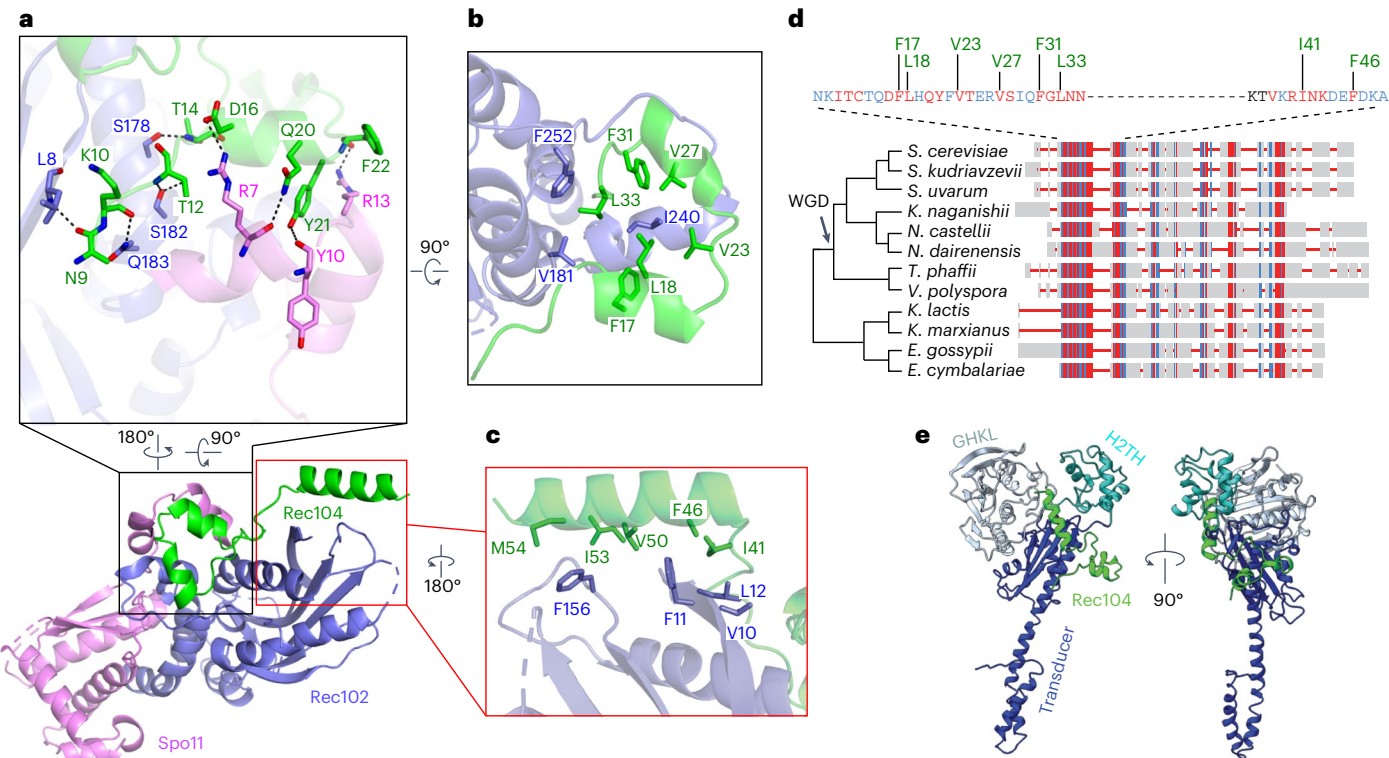

**Fig. 4 | Rec104 structure and interactions with other subunits. a–c,** Details of protein–protein contacts of Rec104 with Rec102 and Spo11: Helix 1 and the preceding residues of Rec104 contact both Rec102 (strand A and the stalk) and Spo11 (N-terminal extension), primarily through backbone and side chain hydrogen-bonding interactions (**a**). Helices 1 and 2 in Rec104, plus the intervening turn, contact the Rec102 stalk and C-terminal extension through hydrophobic interactions (**b**). The third α-helix of Rec104 contacts the Rec102 β-sheet (strands A, B and C) and switch loop through hydrophobic interactions (**c**). The gapped-DNA-bound core complex is shown in **a,b**, while the hairpin-DNA-bound complex is shown in **c** because the resolution was higher for this part of the structure. **d,** Conservation of Rec104 in Saccharomycetaceae. A multiple-sequence alignment of Rec104 sequences from 12 species was generated and visualized using COBALT[45]. Aligned residues are colored on the basis of relative

entropy, with red indicating more conserved residues. The region with the strongest conservation, shown in detail above, matches the portion of Rec104 visible in the cryo-EM structures. Conserved residues mediating hydrophobic interactions are labeled in green. Topology of the cladogram is based on a previous study[46]. WGD, whole-genome duplication. **e,** Position of the structured region of Rec104 relative to the domains of *S. shibatae* Top6B. The core complex was superposed with Topo VI (PDB 2ZBK)[4] by aligning the Cα atoms of strand C from each protein (S147–F156 of Rec102 and M409–T418 of Top6B). Only Rec104 from the core complex is shown (green); the GHKL ATPase domain of Top6B is colored pale cyan, the H2TH domain is colored cyan and the transducer domain is colored dark blue. In the superimposed ensemble, Rec104 wraps around the transducer domain, with helices 1 and 2 lying in front of the stalk and helix 3 lying against the back side of the upper corner of the β-sheet.

and Extended Data Fig. 8f). The equivalent region in Top6B is at the interface with the GHKL domain, as previously noted[14]. However, when we superimposed the core complex on a crystal structure of Top6B (ref. 4), the structured elements of Rec104 were strikingly distinct from the GHKL or HT2H domains of Top6B (Fig. 4e). Although we do not know the disposition of the C-terminal two thirds of Rec104, the end of helix 3 points toward the location of the Top6B GHKL domain (Fig. 4e). Moreover, Rec104 K61, K65, K78 and K79 (which lie just outside the structured segment) crosslink to Rec102 K60, K64 and K79 (ref. 14), all three of which are near the position occupied by the GHKL domain in Top6B.

We conclude that the structured part of Rec104 is not equivalent to any of the domains in Top6B but that the C-terminal part of Rec104 is near or in the GHKL domain position, consistent with the previous proposal about Rec104's location[14]. However, it remains unclear whether the unstructured portion of Rec104 evolved from the GHKL domain or is a wholly unrelated fold that replaced it.

## Protein–DNA contacts and the specificity of DNA binding
An electrostatic surface representation of the core complex identified three positively charged patches that line the DNA-binding channel and mediate non-sequence-specific protein–DNA contacts (Fig. 5a). Patch 1 anchors overhang nucleotide A39 and its adjacent duplex in the pocket between the Spo11 WH and Toprim domains. R131 on the

second α-helix of the WH domain forms a hydrogen bond with the 5′-phosphate of G40, which is the bridging phosphate between the first base of the duplex (G40) and the beginning of the 5′ overhang (A39) (Fig. 5b). This interaction stabilizes the orientation relative to the DNA of the catalytic Y135, which is on the same α-helix (Fig. 5b). In addition, the third α-helix of the WH domain inserts into the major groove, with Q144 interacting with the same phosphate as R131 (Fig. 5b) and R143 interacting with the phosphate backbone of the opposite strand (Fig. 5c). Furthermore, residue K173 at the beginning of the flexible linker connecting the WH and Toprim domains makes one of the rare contacts of Spo11 with a base, forming a hydrogen bond with the N-3 position of G40 (Fig. 5b). This contact is interesting because G is favored at this position in vivo[29]. Toprim domain residues in patch 1 include R266, which forms a hydrogen bond with the phosphate of A42, and E233, which forms a hydrogen bond with the 3′-terminal hydroxyl group of C71 (Fig. 5b).

These DNA interactions explain protein sequence conservation and prior experimental findings. Specifically, R131, Q144, R266 and E233 are invariant in Spo11 and Top6A and K173 is a basic residue in nearly all eukaryotes (Extended Data Fig. 9a). Moreover, the contacts of R131 and Q144 with the 5′-phosphate at the beginning of the duplex and of E233 with the 3′-hydroxyl explain the strong selectivity for a free 3′-OH end and 5′ ssDNA overhang[14]. R131 and E233 are essential for Spo11 function in vivo[7]; substituting K173 to alanine reduced DNA

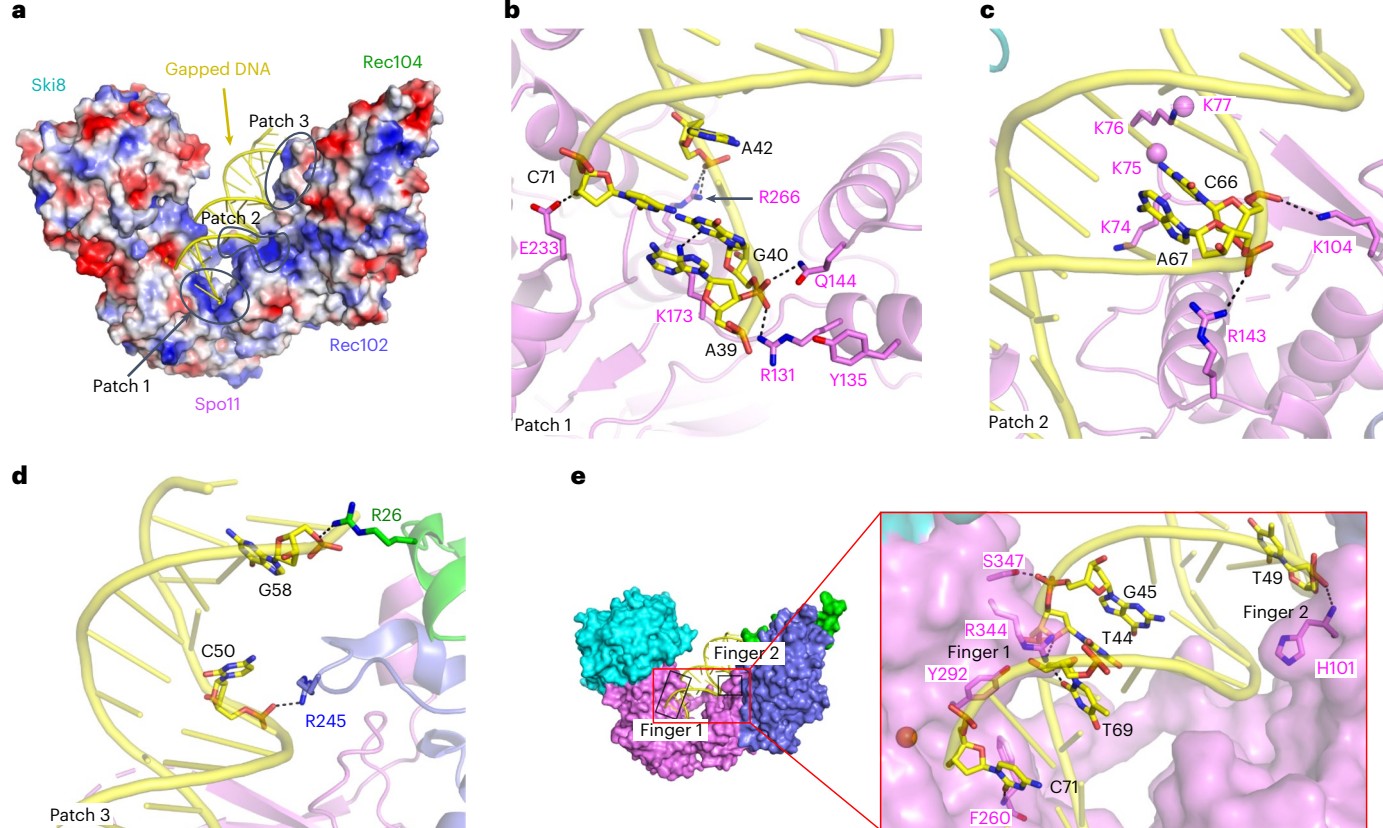

**Fig. 5 | Protein–DNA contacts in the gapped-DNA-bound core complex.** **a**, Electrostatic surface representation of the protein subunits of the Spo11–Ski8–Rec102–Rec104 complex. **b–d**, Details of protein–DNA contacts associated with patch 1 (**b**), patch 2 (**c**) and patch 3 (**d**). Side chain densities that could not be traced are shown as balls. **e**, Details of protein–DNA contacts associated with fingers 1 and 2. In finger 1, R344 forms base-specific hydrogen bonds with the O-4′ of T44 and O-2 of T69, further stabilized by hydrogen bonding of S347 with the phosphate of G45, while the F260 main chain nitrogen also forms a hydrogen bond with the O-2 of base C71.

binding in vitro and delayed and reduced DSB formation in vivo[14], while substituting E233 to alanine reduced DNA binding in vitro ~10-fold[14].

Patch 2 is in the WH domain (Figs. 2a and 5a). The basic K$_{74}$KKK loop surrounds the T63-T64-A65-containing strand, while K104 hydrogen bonds with the phosphate of C66 (Figs. 2a and 5c). The density is weak for the K$_{74}$KKK loop, with K74 and K76 directed toward the DNA backbone. This loop is part of a β-hairpin (the 'wing' in the WH domain) and is cleaved by hydroxyl radicals produced by iron-chelating moieties placed nearby on the DNA[14] but is highly variable in length and sequence in Spo11 proteins (Extended Data Fig. 9a).

Patch 3 is on the outer segment of the DNA-binding channel (Figs. 2a and 5a). It includes Rec102 R245 and Rec104 R26, which form hydrogen bonds with the phosphates of C50 and G58, respectively (Fig. 5d).

We also identified two sets of residues ('fingers') projecting from Spo11 into the minor groove. Finger 1 (F260, Y292 and R344 from the Toprim domain) tracks along the first five base pairs and finger 2 (H101 from the WH domain) forms a main chain amide hydrogen bond with the phosphate of T49 and inserts its imidazole ring into the minor groove (Fig. 5e). F260 is a bulky hydrophobic residue in Spo11 orthologs but glutamine in most Top6A proteins (Extended Data Fig. 9a). The minor groove binding by finger 1 is particularly striking because the base composition bias around Spo11 cleavage sites in vivo suggested a tendency toward a relatively wide and shallow minor groove across precisely this region[29]. An F260A substitution dramatically changes DSB site preference in vivo and reduces DNA-binding affinity in vitro[14], while substituting Y292 to arginine disrupts DSB formation in vivo[7].

## Metal-ion binding

Type IIA topoisomerases (eukaryotic Topo II and bacterial gyrase and Topo IV) are thought to use a two-metal mechanism involving divalent cations bound by acidic residues of the Toprim domain[30–32]. In this model, metal ion A interacts with both bridging and nonbridging oxygens of the scissile phosphate and has a direct role in catalysis, while metal ion B interacts with an adjacent (nonscissile) phosphate and has a structural role in stabilizing protein–DNA interactions (Fig. 6a). Comparatively little is known in type IIB enzymes[3,33].

We observed density in both cryo-EM structures consistent with a single metal ion, most likely Mg$^{2+}$, bound by D288 and D290 of Spo11 (Fig. 6b). When the Spo11 metal-binding pocket was compared with that of *Methanocaldococcus jannaschii* Top6A (ref. 6), the trio of conserved acidic residues was highly congruent but the single Mg$^{2+}$ bound by each protein was in a different position, bound by E197 and D249 in Top6A (equivalent to Spo11 E233 and D288) (Fig. 6b). By comparison to yeast Top2 (ref. 30), we infer that site A is occupied in the Top6A structure, whereas site B is occupied in our Spo11 structure (Fig. 6a,b). In type IIA enzymes, site A has a higher affinity for metal than site B but is thought to rely on the scissile phosphate for two of the coordination contacts[32]. If the same is true for Spo11, the absence of an equivalent of the scissile phosphate at the 3′ end in our structure may explain why Mg$^{2+}$ is not stably bound at site A. The observed binding in site B is consistent with the presumed structural role for the metal in this position. Occupancy of site B is consistently observed in post-cleavage structures of eukaryotic and bacterial topoisomerases[30,34], supporting the conclusion that our structure mimics the postcleavage state.

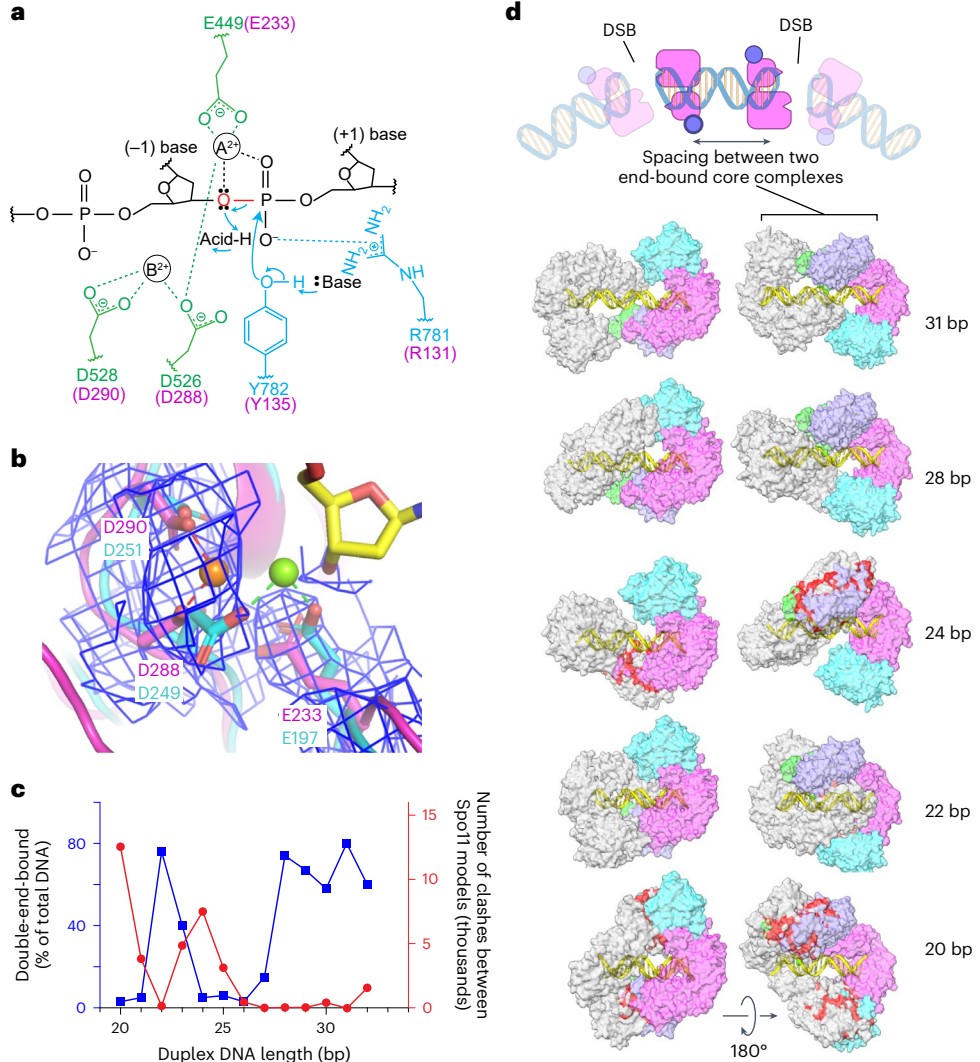

**Fig. 6 | Metal-ion binding and models of higher-order protein–DNA complexes. a**, Proposed two-metal-ion mechanism for catalysis of strand cleavage by type IIA topoisomerases[30]. Amino acids in the active site of *S. cerevisiae* Topo II are indicated, with metal-binding residues in the Toprim domain colored in green and residues from the WH domain colored in blue. Corresponding Spo11 residues are colored in magenta. The general base and acid are unknown. **b**, Superposition of the metal-binding pockets of Spo11 (magenta) and *M. jannaschii* Top6A (cyan; PDB 1D3Y)[6]. Only the metal ion and the side chains of the indicated residues from Top6A are shown. The magnesium in the Spo11 cryo-EM structure is coordinated by D288 and D290, while the magnesium in Top6A is coordinated by D249 and E197, which are equivalent to D288 and E233 in Spo11. The electron density map around the Spo11 triads and Mg²⁺ is shown as a blue mesh; note the lack of detectable density at the position of the

metal (green ball) bound by Top6A. **c,d**, Models for two Spo11 core complexes bound to opposite ends of DNA duplexes of varying lengths. The plot compares the number of clashes between the two core complexes as a function of duplex DNA length (red) with direct measurement of the ability to form double-end-bound complexes in EMSA experiments (blue; data from ref. 35) (**c**). The blue points show the fraction of protein–DNA complexes that have both DNA ends bound. Example models are shown of double-end-bound core complexes with DNA duplexes of 22 bp, 24 bp, 28 bp and 31 bp (lengths do not include the ssDNA overhangs at each end) (**d**). One core complex is colored as in Fig. 1d. The other core complex is colored in gray. Clashes are colored red. The cartoon at the top of **d** illustrates that monomeric core complexes bound to opposite ends of a duplex mimic the back-to-back arrangement of two adjacent Spo11 dimers that have cut the DNA.

These findings provide a framework for understanding the conservation and functional importance of the metal-binding acidic residues in Spo11. E233 and D288 are invariant in Spo11 and Top6A proteins (Extended Data Fig. 9a) and are essential for DSB formation in vivo in yeast[7]. Extrapolating from type IIA enzymes, we propose that the essentiality of both residues traces to them directly coordinating the catalysis-critical Mg²⁺ in site A (Fig. 6a,b). By contrast, D290 is mostly but not strictly conserved (Extended Data Fig. 9a) and it is dispensable for DSB formation in vivo, whereby the substitution with asparagine has little effect on recombination activity in *S. cerevisiae* while the alanine substitution partially decreases DSB formation[7]. It may be that this residue is not essential because it contributes to metal binding only in the noncatalytic site B (Fig. 6a,b).

## Modeling higher-order complexes

**Spacing between adjacent DSBs.** Our cryo-EM structure sheds light on spatial patterns of Spo11–DNA interactions both in vivo and in vitro. Multiple Spo11 complexes can sometimes introduce two or more DSBs close together on the same DNA molecule[35–37]. When such double-cutting occurs, it has a preferred spacing between DSBs with a minimum of ~33 bp (measured from the center of each DSB's 5′ overhang, corresponding to 31 bp of duplex DNA plus the overhangs) and increasing in steps of ~10 bp. The 10-bp periodicity has been proposed to reflect a geometric constraint, in which adjacent Spo11 dimers are co-oriented with their active sites facing in the same direction[35] (Extended Data Fig. 9b). In this model, the observed minimum distance reflects steric constraints that prevent

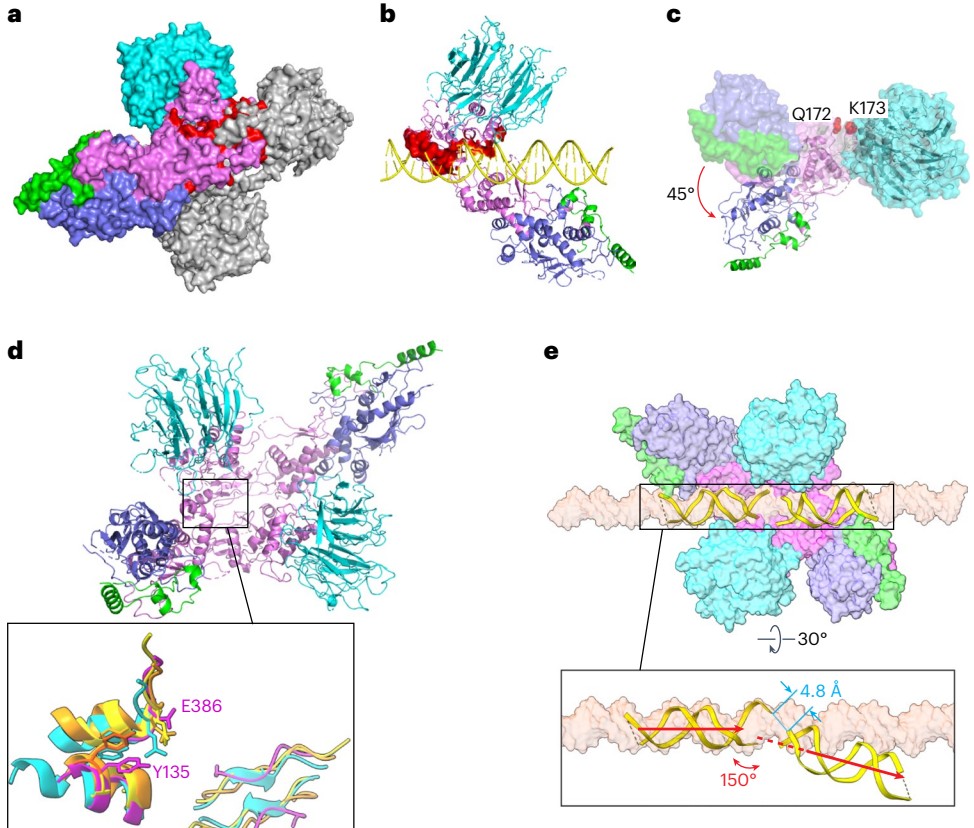

**Fig. 7 | Model of a pre-DSB Spo11 dimer. a**, Predicted Spo11–Spo11 clashes when core complexes are docked together on a B-form DNA. Two copies of the DNA-bound core complexes were aligned onto a B-form DNA using the first two nucleotides of their 5′ overhangs. One Spo11 core complex is colored as Fig. 1d and the other is colored in gray. The protein clashes are colored in red. DNA is not shown. **b**, Predicted clashes between protein and DNA strands. The clashing regions are highlighted in red surface representation. **c**, Rigid-body motion of the Spo11 WH domain relative to the Toprim domain. The Spo11 core complex cryo-EM structure is shown in a surface representation. The ribbon diagram shows the hypothetical configuration of the core complex if the WH and Toprim domains of Spo11 are superposed separately (separated between Q172 and K173; Methods)

onto their cognate domains in *S. shibatae* Top6A from the crystal structure of the Topo VI holoenzyme[4]. **d**, Overview of the model of a Spo11 core complex dimer with inset comparing parts of the Spo11 dimer interface (purple) with the equivalent Top6A dimer interfaces from different species (cyan, *M. jannaschii*; orange, *M. mazei;* yellow, *S. shibatae*.). **e**, Deformation of DNA in the hypothetical pre-DSB dimer model. Proteins of two core complexes are shown in a surface representation and the two DNA duplexes are shown in ribbon diagram with their positions relative to the Spo11 WH domains preserved from the cryo-EM structures. A surface representation of a B-form DNA is included for comparison. The inset (rotated 30°) highlights the deviation from the B-form DNA path in the model.

adjacent co-oriented Spo11 complexes from being fewer than three helical turns apart.

Electrophoretic mobility shift assay (EMSA) experiments with yeast Spo11 core complexes support this interpretation; both 5′ overhang ends of a DNA substrate are readily bound if the duplex DNA segment between the overhangs is ≥28 bp but duplex lengths of 24–27 bp can only be bound at a single end[35] (Fig. 6c). Double-end binding by monomeric core complexes in vitro likely mimics how close two adjacent Spo11 dimers can be on an intact DNA molecule before cleavage (Fig. 6d, top). Interestingly, reducing the duplex length even further (to 22–23 bp) allows core complexes to again bind at both ends[35] (Fig. 6c). These results suggest that steric clashes that preclude double-end binding at 24–27 bp are relieved if the two DNA ends are rotated relative to one another. Different orientations of adjacent Spo11 complexes are possible with free proteins in vitro but not with the constrained complexes presumed to be present in vivo (Extended Data Fig. 9b).

These patterns of DNA cleavage in vivo and DNA binding in vitro are well explained by the extensive left-handed wrap of the core complex around the DNA in the cryo-EM structure. Two core complexes could be modeled on each end of a 28-bp DNA duplex with essentially no steric clashes but modeling on a shorter DNA segment (24 bp) resulted in

substantial overlap between the Rec102 and Rec104 moieties of the two core complexes (Fig. 6c,d). Consistent with EMSA data, the clashes could be almost entirely resolved by shortening the DNA still further to 22 bp, which rotates the core complexes relative to one another and allows them to interdigitate (Fig. 6c,d). Moreover, a 31-bp duplex is the shortest distance that can accommodate a pair of core complexes that is both co-oriented and lacking in steric clashes (Fig. 6d). This agrees well with in vivo spacing because a 31-bp duplex corresponds to a center-to-center DSB distance of 33 bp (that is, the minimum preferred distance for double cuts)[35,37].

**Model of a pre-DSB Spo11 dimer.** We attempted to model a catalytically competent Spo11 dimer by docking two copies of the monomeric DNA-bound core complex together. If the two DNA copies are aligned to make a roughly B-form duplex, the WH domain from each Spo11 clashes sterically with the Toprim domain from the other (Fig. 7a). In addition, the nonoverhang strand from one complex clashes with the C-terminal part of α-helix 8 of Spo11 from the other complex and the overhang strand collides with a loop near finger 1 (Fig. 7b). Thus, the configurations of both protein and DNA in the cryo-EM structure are incompatible with a plausible structure of a pre-DSB complex on B-form DNA.

Because Spo11 core complexes are flexible in solution[14], we asked whether simple rigid-body motions of their domains could resolve the clashes. To answer this, we separately aligned the Spo11 WH domain (together with DNA, Rec102 and Rec104) and the Toprim domain (together with Ski8) to the cognate domains of Top6A in the *Saccharolobus shibatae* Topo VI dimer structure[4] (Extended Data Fig. 9c) because the segment between the domains is thought to be a flexible linker[6,14]. This alignment rotates the WH domain 45° relative to the Toprim domain and, notably, eliminates all of the steric clashes (Fig. 7c). The model also matches segments of Spo11 well with the two major Top6A–Top6A interfaces in Topo VI dimer structures[4–6]: a pseudocontinuous β-sheet on the underside of the Toprim domain away from the DNA (K206 to P211 from one Spo11 interacting with its match on the other Spo11) and an interface between the WH domain of one Spo11 near the catalytic tyrosine and a C-terminal region of the other Spo11 that includes an invariant residue E386 that is brought in close proximity to the catalytic tyrosine (Fig. 7d and Extended Data Fig. 9a). This model is, thus, a plausible representation of a dimeric pre-DSB complex. We further speculate that DSB formation is accompanied by a Spo11 conformation change into a postcleavage state resembling our cryo-EM structures and that the steric clashes described above may prevent DNA end religation, rendering DSB formation irreversible.

The model predicts an intriguing deformation of the DNA. We preserved the relationship of the DNA to the WH domain because this also preserves the large majority of the protein–DNA contacts from the cryo-EM structure. In doing so, the overhang ends of the two DNA segments come into close proximity (~4.8 Å separating the inferred positions of 5′ and 3′ ends of each strand) but make a V shape with a 150° angle and with the helical axes slightly offset by ~5 Å (Fig. 7e). Empirical evidence for DNA deformation by Spo11 comes from the 130° angle of the DNA in our gapped double-hairpin structure (Fig. 2d) and by previous DNA-binding experiments that revealed bent DNA by atomic force microscopy and apparent preferential binding to bent duplex DNA (inferred from higher affinity for binding 100-bp versus 400-bp minicircles)[14]. Topo VI and other type II topoisomerases bend DNA before cleavage[6,14,38]; hence, it is likely that Spo11 core complexes may do so as well. Our model, thus, provides a framework for understanding the structural determinants of DNA bending, for which no high-resolution structural information is currently available for any dimeric Topo VI or Spo11 complex.

## Conclusions

Ever since the discovery of the DSB-forming activity of Spo11 more than a quarter-century ago[39,40], there has been a lack of empirical structures of Spo11 and its accessory proteins. Our structures provide insight into the architectures of the individual subunits and of their interfaces with each other and DNA. We confirm the structure and position of Ski8, uncover unexpected differences between Rec102 and archaeal Top6B proteins and partially resolve outstanding questions about the structure and spatial disposition of Rec104. Our findings further explain the high affinity and exquisite selectivity of core complex binding to DNA ends that have a recessed 3′-OH. Lastly, the structures provide a molecular framework to understand nonrandom DSB patterns in vivo, including the biased base composition around cleavage sites and the spacing of double cuts.

It remains unclear why yeast core complexes are thus far not competent to cleave DNA in vitro. Perhaps the high affinity for DNA ends precludes dimer assembly on DNA when ends are available. A related issue is that the high-affinity DNA-binding state we observe has the proteins in a conformation that is not compatible with dimer formation. The ability of Spo11 accessory proteins Rec114, Mei4 and Mer2 to recruit Spo11 core complexes and present them in a coherent orientation (Extended Data Fig. 9b) may facilitate formation of DSB-competent dimers[41–44]. Inclusion of these other proteins may, thus, be needed for DNA cleavage in vitro. Alternatively, it may be possible to overcome the barrier to dimer formation by using DNA substrates that disfavor the monomer configuration. In any case, our results point to the Spo11 monomer–dimer dynamic as being key to controlling the DSB-forming activity.

## Online content

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

## Methods

### Purification of *S. cerevisiae* Spo11 core complexes

The expression plasmids and baculoviruses were prepared as described previously[14]. The Spo11 core complex was expressed by coinfecting Gibco *Spodoptera frugiperda* Sf9 cells (Thermo Fisher 11496015) with a combination of viruses at a multiplicity of infection of 2.5 each as described previously[14]. Typically, 500 ml of suspension culture was collected 62 h after infection. The cells were lysed by sonication in 25 mM HEPES-NaOH pH 7.4, 500 mM NaCl, 0.1 mM DTT, 20 mM imidazole, 1× cOmplete protease inhibitor tablet (Roche) and 0.1 mM PMSF and centrifuged at 43,000$g$ for 30 min. Cleared extract was loaded onto 1 ml of Ni-NTA resin (Qiagen) pre-equilibrated with nickel buffer (25 mM HEPES-NaOH pH 7.4, 500 mM NaCl, 10% glycerol, 0.1 mM DTT, 20 mM imidazole and 0.1 mM PMSF) and then washed extensively with nickel buffer. Core complexes were eluted in nickel buffer containing 250 mM imidazole and then further purified on anti-Flag M2 affinity resin (Sigma). To do so, fractions from the Ni-NTA elution containing protein were pooled and diluted in three volumes of Flag buffer (25 mM HEPES-NaOH pH 7.4, 500 mM NaCl, 10% glycerol and 1 mM EDTA) before binding to the anti-Flag resin. Core complexes were eluted from anti-Flag resin with Flag buffer containing 250 µg ml$^{-1}$ 3xFlag peptide (Sigma). Fractions containing protein were pooled and loaded on a Superdex 200 Increase 10/300 GL column (Cytiva) pre-equilibrated in 25 mM HEPES-NaOH pH 7.4, 300 mM NaCl, 5 mM EDTA and 2 mM DTT. Fractions containing protein were concentrated in 50-kDa-cutoff Amicon centrifugal filters (Millipore). Aliquots were frozen in liquid nitrogen and stored at −80 °C.

### Cryo-EM structure determination

Purified core complexes were diluted to a final concentration of ~0.2 mg ml$^{-1}$ and incubated with the hairpin (5′-TAGCAATGTAATCGT CTATGACGTTAACGTCATAGACGATTACATTGC) or gapped DNA (5′-TAGGCCGTCGGCTACTAAAAGTAGCCGACGGCCGGATTAGCA ATGTAATCGTCTTAAGACGATTACATTGC) substrates at a molar ratio of 1:2 (protein:DNA) in binding buffer (25 mM HEPES-NaOH pH 7.5, 150 mM NaCl, 5 mM MgCl$_2$, 2 mM DTT and 2% glycerol) for 1 h on ice. An aliquot of the incubated mixture (3.5 µl) was applied onto glow-discharged UltrAuFoil 300 mesh R 1.2/1.3 grids (Quantifoil) at -4 °C. Grids were blotted for 2 s at 100% humidity at 4 °C and flash-frozen in liquid ethane using a Vitrobot Mark IV (Thermo Fisher Scientific). The twofold molar excess of DNA was expected to favor singly-bound DNA molecules but would allow recovery of doubly-bound molecules if the affinity for binding by the second core complex were comparable to binding by the first. Because binding to the hairpin end is much lower affinity than binding to the end with the 5′ overhang, only singly-bound complexes were expected for the hairpin substrate. However, we did not know a priori whether a dimer of core complexes might be able to bind to the gapped substrate.

Images were collected on a Titan Krios G2 (FEI) transmission EM instrument operating at 300 kV with a K3 direct detector (Gatan) using a 1.064-Å pixel size at the Memorial Sloan Kettering (MSK) Cancer Center. The defocus range was set from −1.0 to −2.5 µm. Videos were recorded in super-resolution mode at an electron dose rate of 20 e− per pixel per s with a total exposure time of 3 s and intermediate frames were recorded every 0.075 s for an accumulated electron dose of 53.00 e$^{-}$ per Å$^2$.

The protocol for cryo-EM reconstruction is presented in Extended Data Figs. 2a–d and 3a–d. The super-resolution movies (0.532 Å per pixel) were motion-corrected and Fourier-cropped twice using Motion-Cor2 (ref. 47). Contrast transfer function parameters were estimated by CTFFIND-4 (ref. 48). All other steps of image processing were performed by RELION 3.0 (ref. 49) and Cryosparc version 3.3.0 (ref. 50).

For hairpin-bound core complexes, after blob picking from 2,338 images without reference, a total of 1,829,931 particles extracted after three-pixel binning were applied to two rounds of two-dimensional (2D) classification. All particles were applied to multiple rounds of heterogeneous refinement by using the 4.8-Å-resolution model

generated from RELION 3.0. The final 128,787 particles were polished and yielded a reconstruction EM map with an average resolution of 3.66 Å. All reported map resolutions are from gold-standard refinement procedures with the Fourier shell correlation (FSC) cutoff being 0.143 (Extended Data Fig. 2).

For gapped-DNA-bound core complexes, after blob picking from 5,521 images without reference, a total of 7,804,373 particles extracted after four-pixel binning were applied to the first round of 2D classification. A total of 6,930,069 particles selected from 2D classification were applied to multiple rounds of three-dimensional (3D) classification. One of the 3D classes with good secondary-structure features and the corresponding 1,314,980 particles were re-extracted without binning. The selected particles were imported to new initial models and hetero refinement and 548,674 particles were selected from the best class for homogeneous refinement. After nonuniform refinement and local refinement, we obtained an EM map with an average resolution of 3.29 Å. All reported map resolutions are from gold-standard refinement procedures with the FSC cutoff being 0.143 (Extended Data Fig. 3).

Spo11, Ski8 and Rec102 models were downloaded from the Alpha-Fold2 database[10] and fitted into the cryo-EM density map as rigid bodies. The DNA model was generated using Coot[51]. We also generated a model of the Rec102–Rec104 subcomplex using ColabFold[52] and fitted this model into the cryo-EM density map as well. The side chains of the visible N-terminal portion of Rec104 could be traced in the density map of the hairpin-bound core complex. Atomic coordinates were refined against the map by real-space refinement in PHENIX[53] by applying geometric and secondary-structure restraints. Details of data collection, image processing and model building are listed in Table 1 and shown in Extended Data Figs. 2 and 3. All figures were prepared using PyMol (https://pymol.org/2/), UCSF Chimera[54] or UCSF ChimeraX[55].

### DNA substrates and EMSAs

EMSAs used the same DNA substrates as in the cryo-EM structures. Both the hairpin and the gapped substrates were assembled by self-annealing in annealing buffer (100 mM NaCl, 10 mM Tris-HCl pH 8 and 1 mM EDTA). Substrates were 5′-end-labeled with [γ-$^{32}$P]ATP (Perkin Elmer) and T4 polynucleotide kinase (New England Biolabs) and purified by native PAGE. Binding reactions (10 µl) were carried out in 25 mM Tris-HCl pH 7.5, 7.5% glycerol, 100 mM NaCl, 2 mM DTT, 5 mM MgCl$_2$ and 1 mg ml$^{-1}$ BSA with 0.1 nM DNA. Complexes were assembled for 30 min at 30 °C and separated on 6% polyacrylamide DNA retardation gels (Invitrogen). Gels were dried and analyzed by phosphor imaging (Fuji).

### Modeling higher-order assemblies of Spo11 core complexes

All modeling was performed in ChimeraX[56]. The Spo11 double-end binding complex was modeled by aligning the DNA from a Spo11 core complex structure to each end of a 34-bp B-form DNA. To generate models with different lengths of DNA, the core complex at one end was fixed, while the other Spo11 core complex was moved along the B-form DNA path by the specified number of base pairs.

To model a potential pre-DSB assembly containing two core complexes bound to B-form DNA, the individual DNA sequences in two copies of the cryo-EM structure were each aligned to a B-form DNA segment such that the first nucleotide in the 5′ overhang from one DNA sequence was one nucleotide away from the 3′ end of the other DNA sequence.

To model a pre-DSB dimer based on a Topo VI dimer, the cryo-EM structure was separated into two parts; one included residues 1–172 of Spo11 plus Rec102 and Rec104, while the other included the remaining C-terminal part of Spo11 plus Ski8. The two parts were aligned separately to the cognate domains of the *S. shibatae* Top6A subunit from a crystal structure of the Topo VI holoenzyme (Protein Data Bank (PDB) 2ZBK)[4]. The alignment of N-terminal part of Spo11 resulted in an r.m.s.d. of ~1.2 Å over 32 atom pairs; alignment of the C-terminal part of Spo11 resulted in an r.m.s.d. of ~1.1 Å over 70 atom pairs. The session files are available from Mendeley (https://doi.org/10.17632/dx3hx827fp.1).

## Yeast strains and targeting vectors

All yeast strains were from the SK1 background (Supplementary Table 1). All vectors were generated on the basis of pSK275, pSK276, pSK282, pSK293, pSK305 and pSK310 for both yeast two-hybrid assays and heteroallele recombination assays (Supplementary Table 2).

## Yeast two-hybrid assays

Y2H vectors were transformed separately into haploid strains SKY661 and SKY662 and selected on appropriate synthetic dropout medium. Strains were mated and streaked for single-diploid colonies on medium lacking tryptophan and leucine. Single colonies were grown overnight in selective medium containing 2% glucose. Cells were lysed and a quantitative β-galactosidase assay was performed using the yeast β-galactosidase assay kit following the manufacturer's protocols (Thermo Scientific). For Y2H experiments in meiotic conditions, after overnight culture in selective medium with glucose, cultures were washed twice then incubated with 2% potassium acetate for 20 h to induce meiosis.

Y2H assays do not necessarily reflect only those interactions that are direct, as the interactions may be indirect or supported by other proteins. Interactions of Spo11 with either Rec102 (Fig. 3d) or Rec104 (Extended Data Fig. 8c) were measured in meiotic conditions because these interactions require the presence of other members of the core complex (for example, interaction of Spo11 with Rec102 requires Rec104 to also be present)[23]. Interaction of Rec102 with Rec104 (Extended Data Fig. 8e) was measured in vegetatively growing reporter cells.

For immunoblotting, yeast cells expressing the Y2H protein variants were cultured overnight in selective medium. Equivalent cell amounts based on optical density measurements at 600 nm were harvested and treated with 0.2 M NaOH and then proteins were solubilized in 100 µl of 2× Laemmli sample buffer (Bio-Rad). A 10-µl aliquot of each sample was run on a 4–12% Bis-Tris PAGE gel (Invitrogen) and transferred to a 0.2-µm PVDF membrane (Thermo Scientific). The membranes were blocked using StartingBlock Blocking Buffer (Thermo Scientific) for 20 min at room temperature and probed with primary antibodies to LexA (rabbit, 1:2,000, Sigma-Aldrich), hemagluttinin (HA; rat, horseradish peroxidase (HRP)-conjugated, 1:1,000, Roche) and α-tubulin (rat, 1:2,000, Invitrogen, for Rec104 blots; rat, HRP-conjugated, 1:2,000, Santa Cruz, for Spo11 and Rec102 blots). Following three 10-min washes with PBS-T at room temperature, the membranes were developed using an enhanced chemiluminescence western blotting substrate (Thermo Scientific) and imaged with a ChemiDoc imaging system for HRP-conjugated primary antibodies. Alternatively, membranes were incubated for 1 h at room temperature with fluorophore-conjugated species-specific secondary antibodies (donkey antirabbit IgG (Alexa Fluor 488, 1:5,000, Invitrogen) or donkey antirat IgG (Alexa Fluor 594, 1:5,000, Invitrogen)); then, the membranes were washed three times with PBS-T for 10 min each and imaged using a ChemiDoc imaging system.

## Heteroallele recombination assays

Heteroallele recombination assays were performed in diploid strains heterozygous for the *arg4-Bgl* and *arg4-Nsp* alleles[57] along with homozygous deletion mutations for *spo11*, *rec102* or *rec104*. Complementation test plasmids expressing wild-type or mutant versions of Spo11, Rec102 or Rec104 (Supplementary Table 2) were introduced by lithium acetate transformation. The Y2H fusion constructs were used for this purpose. Three independent colonies were cultured overnight on synthetic complete medium lacking tryptophan and leucine to select for the test plasmid and then transferred to YPA (1% yeast extract, 2% peptone and 1% potassium acetate) for 13.5 h before meiotic induction in 2% potassium acetate. After at least 3 h in meiosis, appropriate dilutions of cultures were plated on synthetic complete medium lacking arginine to measure the frequency of arginine-positive recombinants and on YPD (1% yeast extract, 2% peptone and 2% dextrose) to measure colony-forming units.

## Phylogenetic analyses

Protein-coding sequences of Rec104 from different fungal species were identified using the National Center for Biotechnology Information (NCBI) basic local alignment search tool (BLAST). Conserved regions were identified using a subset of 12 species and visualized using COBALT[45]. For the analysis of structural conservation in Rec102 and Top6BL homologs, AlphaFold2 structure predictions were downloaded from the AlphaFold2 database[58]. A BLAST search for a Rec102 homolog in *Sordaria macrospora* identified a predicted structure (AF-F7VPQ4-F1, NCBI XP_003350816.1) that lacks one of the conserved β-strands and the characteristic WKxY motif. Inspection of the genomic locus suggests that a splice site is misannotated; correcting this results in a longer second exon, adding the missing sequence. We used the new sequence to model the structure using ColabFold[52].

## Reporting summary

Further information on research design is available in the Nature Portfolio Reporting Summary linked to this article.

## Data availability

The atomic coordinates and cryo-EM density maps for the core complex bound to hairpin DNA (PDB 8URU and EMD-42501) or gapped DNA (PDB 8URQ and EMD-42497) were deposited to the Research Collaboratory for Structural Bioinformatics PDB and EM Data Bank, respectively. We used the following published atomic coordinate accessions: PDB 2ZBK, PDB 2Q2E, PDB 1D3Y, AF-Q02721-F1, AF-P33323-F1 and AF-P23179-F1. Source data are provided with this paper.

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

## Acknowledgements

This article is subject to the Open Access to Publications policy of the Howard Hughes Medical Institute (HHMI). HHMI lab heads previously granted a nonexclusive CC BY 4.0 license to the public and a sublicensable license to HHMI in their research articles. Pursuant to those licenses, the author-accepted manuscript of this article can be made freely available under a CC BY 4.0 license immediately upon publication. We thank members of the Keeney and Patel laboratories for discussions and experimental advice and J. De La Cruz (MSK) for support in cryo-EM data collection. MSK core facilities are supported by National Cancer Institute Cancer Center support grant P30 CA08748. K.L. was supported in part by the Damon Runyon Cancer Research Foundation (DRG-[2389-20]). M.A. was supported in part by an EMBO long-term fellowship (ALTF 905-2019). This work was supported by National Institutes of Health (NIH) grant R01 HD110120 (to S.K. and D.J.P.), an MSK Basic Research Innovation Award (BRIA; to S.K. and D.J.P.), a postdoctoral BRIA (to M.A.), a Leukemia and Lymphoma SCOR 7021020 grant (to D.J.P.), the Maloris Foundation (to D.J.P.) and NIH grant R35 GM118092 (to S.K.). S.K. is an HHMI investigator. The funders had no role in study design, data collection and analysis, decision to publish or preparation of the manuscript.

## Author contributions

C.C.B., S.K. and D.J.P. conceptualized the project. Y.Y., J.W., K.L., Z.Z., M.A., C.C.B., S.K. and D.J.P. designed the experiments. Y.Y., J.W., K.L., Z.Z., M.A., C.C.B. and S.P. performed the experiments. Y.Y., J.W., K.L., Z.Z., M.A., C.C.B., S.K. and D.J.P. analyzed the data. S.K. and D.J.P. supervised the research. K.L., M.A., S.K. and D.J.P. secured funding. Y.Y., J.W., K.L., S.K. and D.J.P. wrote the paper with input from Z.Z. and M.A. All authors edited the manuscript.

## Competing interests

The authors declare no competing interests.

## Additional information

**Extended data** is available for this paper at https://doi.org/10.1038/s41594-024-01382-8.

**Correspondence and requests for materials** should be addressed to Dinshaw J. Patel or Scott Keeney.

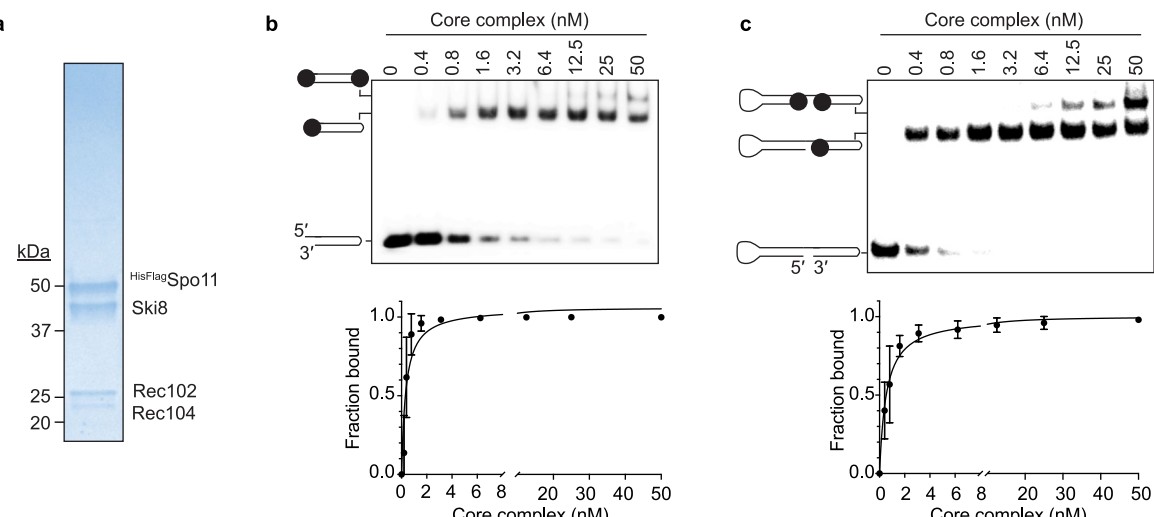

**Extended Data Fig. 1 | Spo11 core complex purification and EMSA assays.**
**a**, SDS-PAGE gel of purified core complex ( ~ 1 µg), stained with Coomassie. The image is representative of results from > 6 independent preparations. **b,c**, EMSA assays and quantification of core complex binding to the hairpin DNA substrate (**b**) and the gapped DNA substrate (**c**). In panel **c**, the position of the second core complex that gives the slowest migrating band is unknown; it could be at the nick location as shown or it could associate with one of the hairpin ends. Error bars indicate mean ± SD of three replicates.

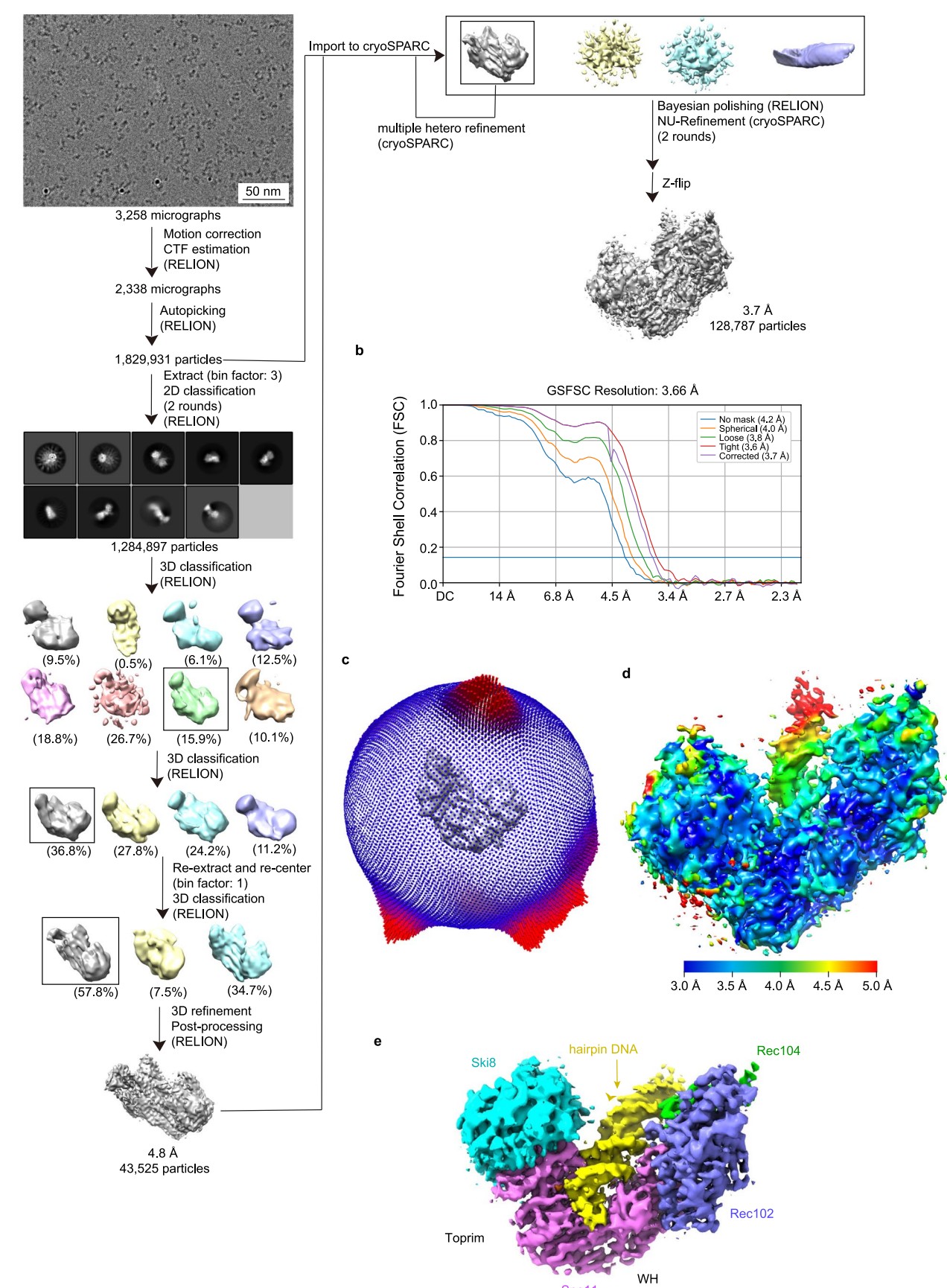

**Extended Data Fig. 2 | Cryo-EM reconstruction of the core complex bound to hairpin DNA. a,** Flow chart of cryo-EM image processing. **b,** Global Fourier shell correlation (FSC) curves. The overall cryo-EM map resolution is 3.7 Å with FSC set at 0.143. **c,d,** Euler angle distribution (panel **c**) and final 3D reconstructed map colored according to local resolution (panel **d**). **e,** Density map.

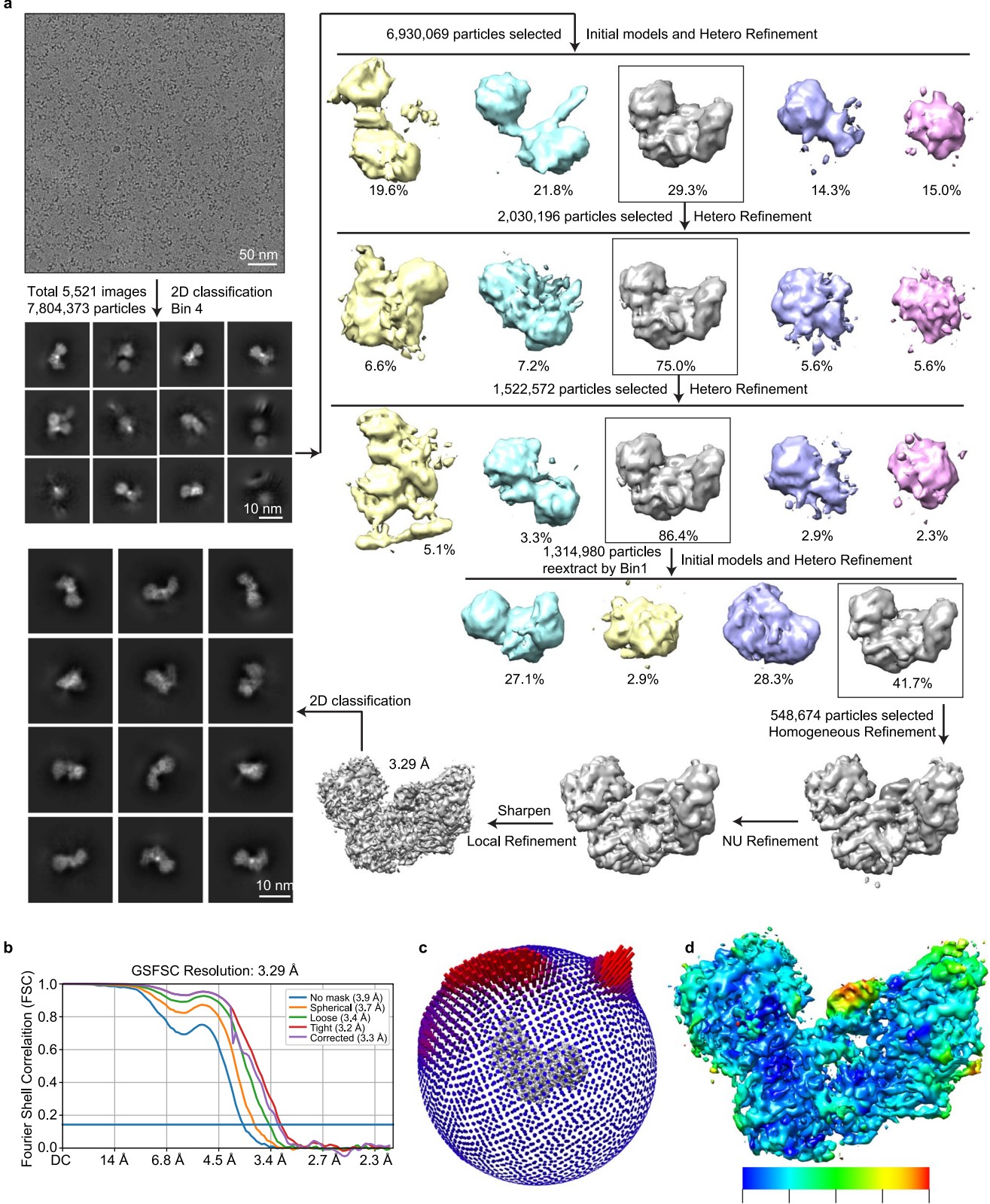

**Extended Data Fig. 3 | Cryo-EM reconstruction of the core complex bound to gapped DNA. a**, Flow chart of cryo-EM image processing. **b**, Global FSC curves. The overall cryo-EM map resolution is 3.3 Å with FSC set at 0.143. **c**,**d** Euler angle distribution (panel **c**) and final 3D reconstructed map colored according to local resolution (panel **d**).

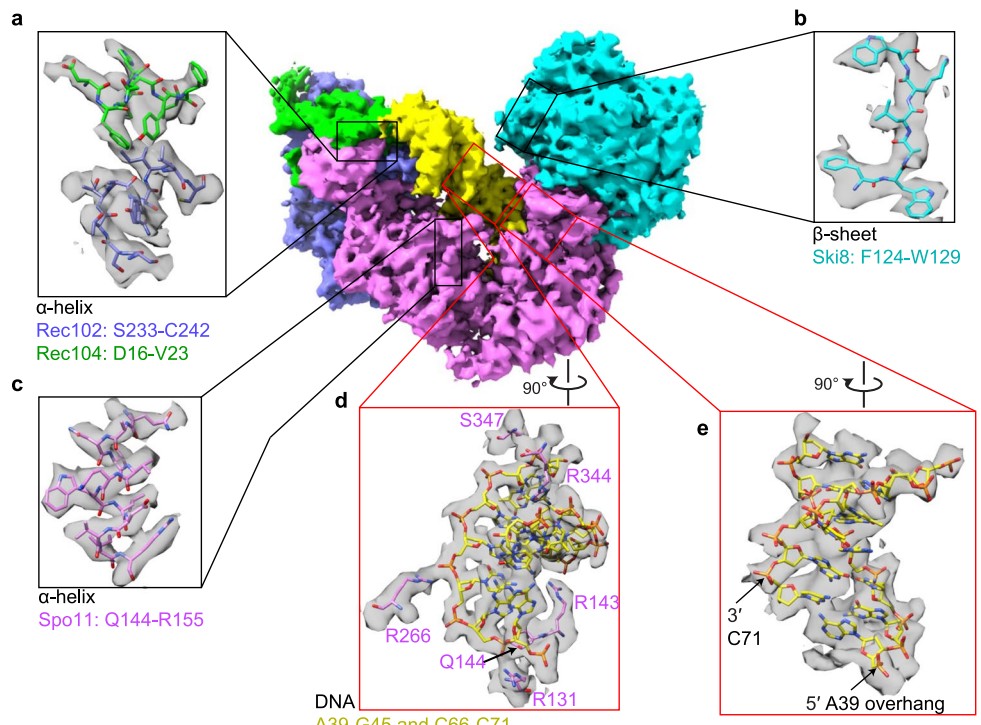

**Extended Data Fig. 4 | Examples of protein side chain and DNA base-sugar-phosphate identification in the cryo-EM structure of the core complex bound to gapped DNA. a–c**, Examples of fitting of protein amino acid side chains into the density map for interacting helices of Rec102 and Rec104 (panel **a**), a segment of β sheet from Ski8 (panel **b**), and a segment of α helix from Spo11 (panel **c**). **d**, Example of fitting DNA base-sugar-phosphate and interacting protein side chains into the density map for Spo11 interaction with the DNA. **e**, Example of fitting DNA base-sugar-phosphate into the density map, highlighting the 3′-OH and first nucleotide of the 5′ overhang.

**a** Spo11

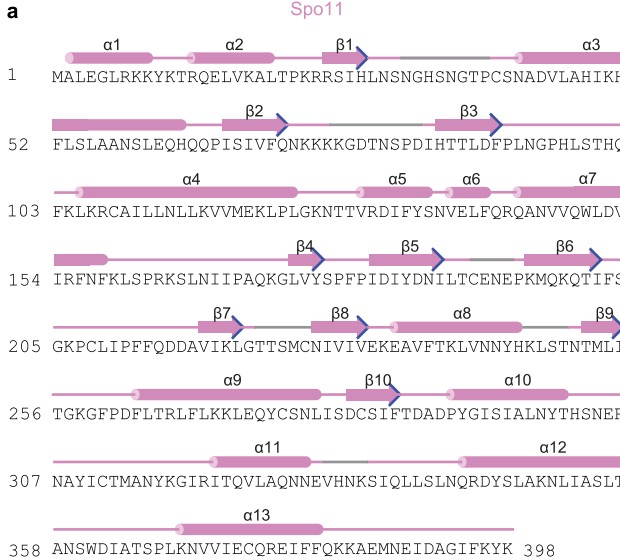

**b** Ski8

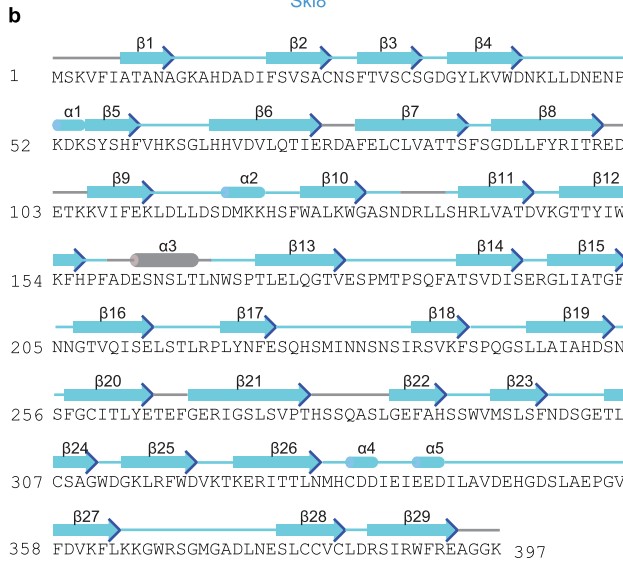

**c** Rec102

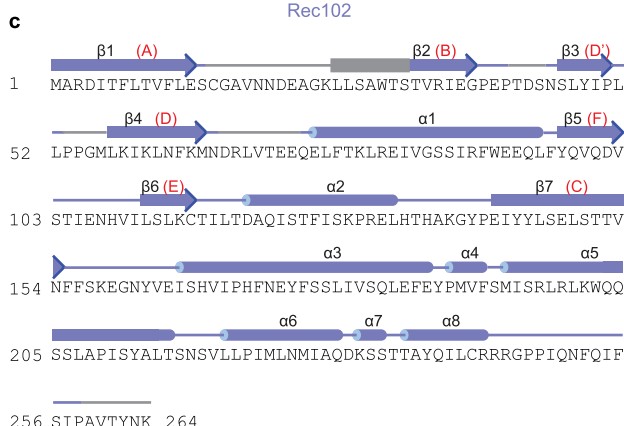

**d** Rec104

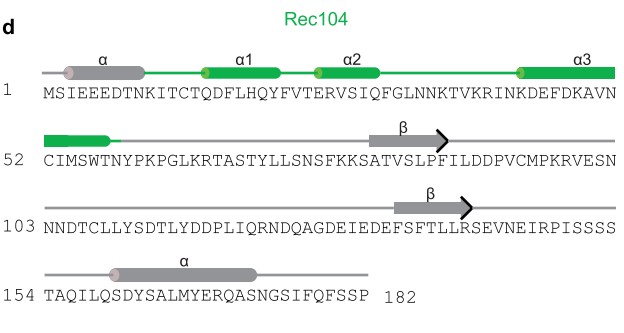

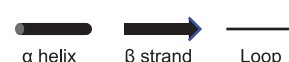

**Extended Data Fig. 5 | Protein secondary structures. a–d**, The secondary structures for Spo11 (panel **a**), Ski8 (panel **b**), Rec102 (panel **c**), and Rec104 (panel **d**) were determined from the cryo-EM structure of the core complex bound to gapped DNA. Gray-colored regions are not visible in the cryo-EM structure. Red letters in parentheses in panel **c** indicate the conserved strand designations shown in Fig. 3b.

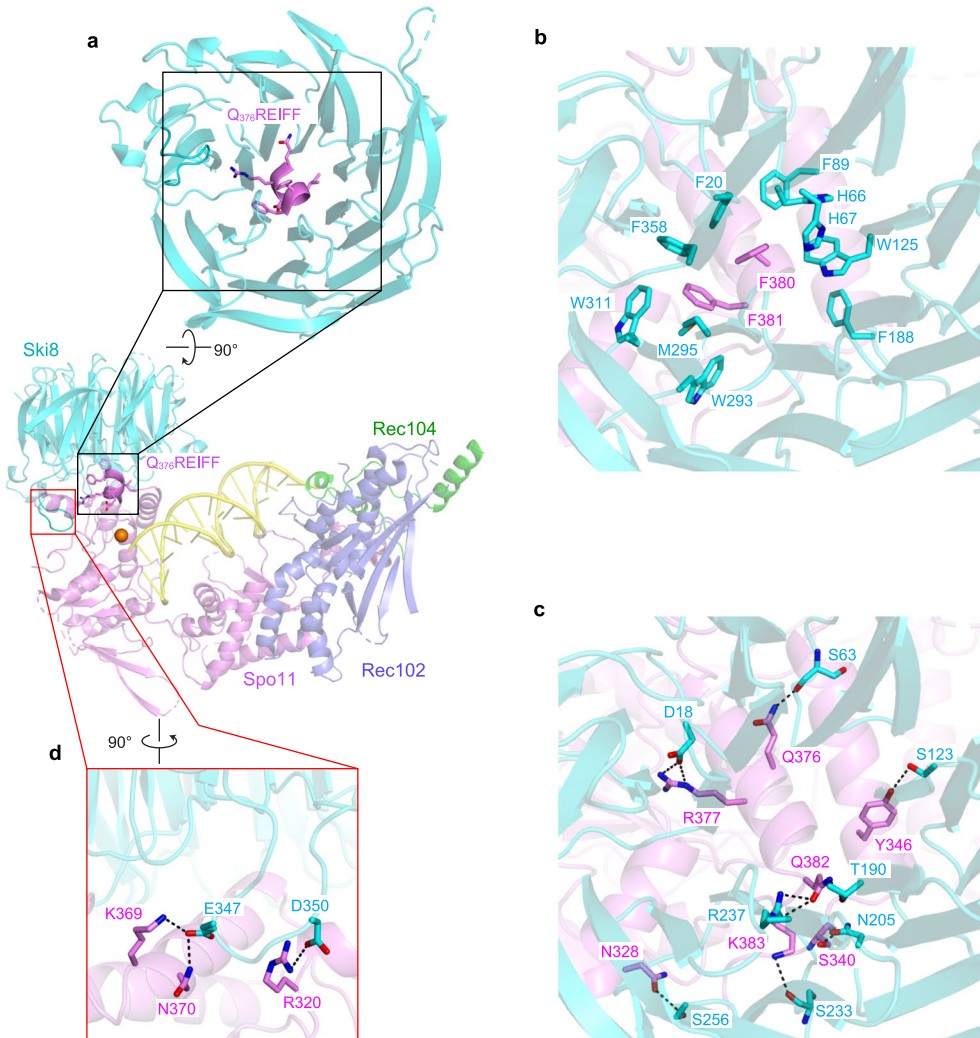

**Extended Data Fig. 6 | Interactions between Ski8 and Spo11.** All images are from the cryo-EM structure of the core complex bound to gapped DNA. **a**, QREIFF motif from Spo11 (magenta) and the WD40 repeats from Ski8 (cyan). Ski8 in the cryo-EM structures adopts a propeller folding topology formed by seven WD40 repeats and recognizes the sequence $Q_{376}$REIFF in the Spo11 Toprim domain. **b**, Hydrophobic interactions between Ski8 WD40 repeats and Spo11. The hydrophobic core of the interface is formed by Spo11 residues F380 and F381 and surrounding Ski8 aromatic residues involved in CH-π and π-π stacking interactions. **c**, Hydrogen bond interactions between Ski8 and Spo11. A hydrogen bonding network surrounds the hydrophobic core illustrated in panel **b**, including bonds between Q376 and R377 of Spo11 and Ski8 residues S63 and D18, respectively. **d**, Additional hydrogen bonding interactions between Spo11 and the extended loop in Ski8. Computational prediction had suggested that an extended loop in Ski8 expands the interface with Spo11[9]. In agreement, we observed additional hydrogen bonding interactions that further stabilize the interface.

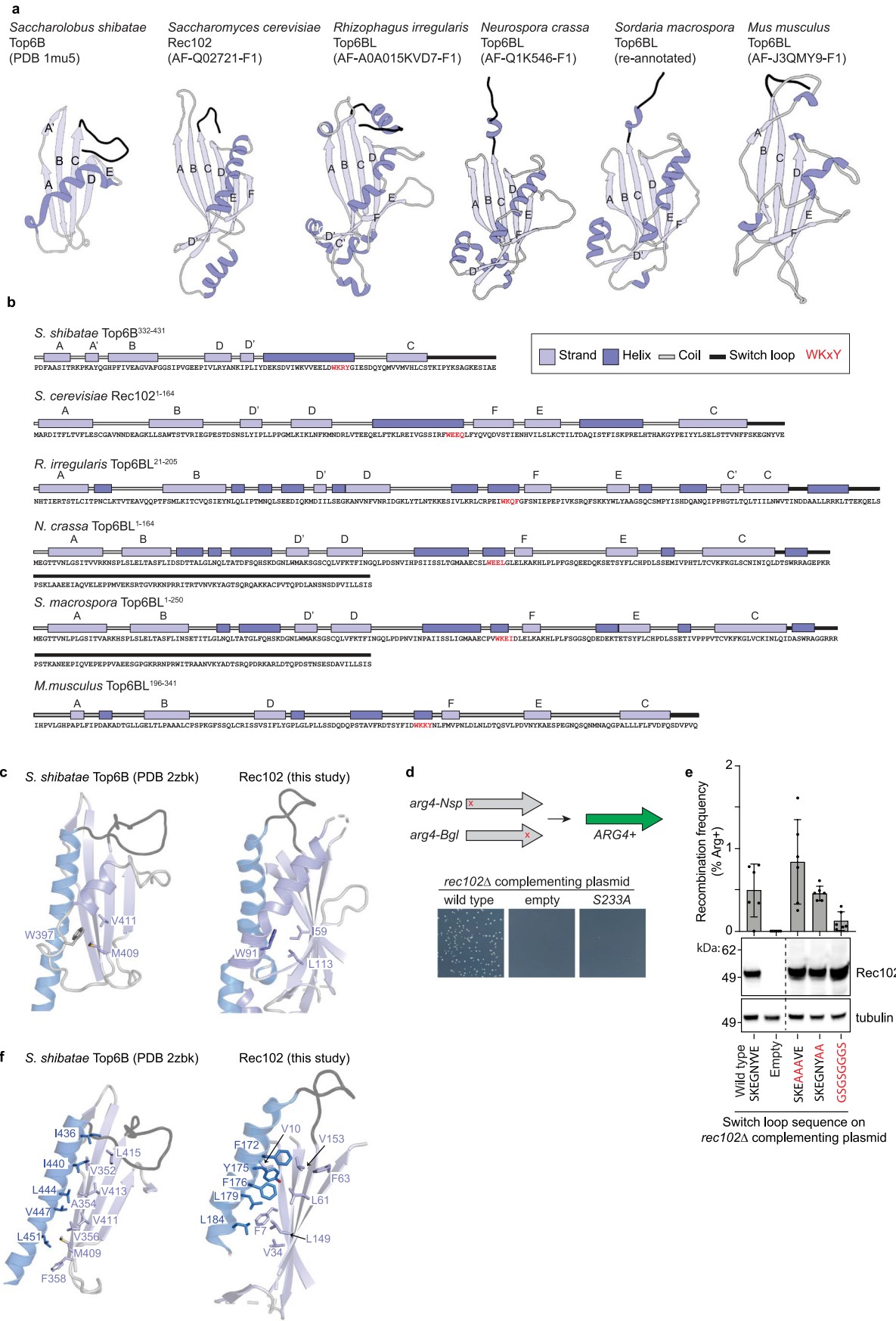

**Extended Data Fig. 7 | See next page for caption.**

**Extended Data Fig. 7 | Sequence and structural conservation of Rec102 and Top6B. a**, Top6B and Rec102/Top6BL architectures. Strands, helices, and switch loops are colored as in Fig. 3b. All structures show truncated proteins, starting at the first β strand (strand A) and ending with the switch loop before the start of the stalk. The switch loop is truncated for the *N. crassa* and *S. macrospora* proteins for clarity. **b**, Secondary structure and sequence information for models shown in **a**. The conserved WKxY motif is highlighted in red and β strands are labeled as in **a. c**, Packing for *S. shibatae* Top6B (PDB: 2zbk) in comparison to Rec102 for the W in the WKxY motif. Ribbons and sticks are colored as in Fig. 3b. **d**, Cartoon illustrating the heteroallele recombination assay. In a strain heterozygous for two different mutant versions of *arg4* (gray arrows with mutations marked with red x's), meiotic recombination can generate a functional copy of *ARG4* (green arrow) by gene conversion and/or crossing over. Below are representative images of Arg+ prototrophic colonies from a *rec102Δ* strain carrying complementing

plasmids expressing the indicated version of Rec102. **e**, Effects of switch-loop mutations on Rec102 activity in vivo. LexA-Rec102 fusion proteins with the indicated switch-loop sequence were used to complement a *rec102Δ* mutation. The wild-type switch-loop sequence is shown in black and the amino acid substitutions are in red. The graph shows the frequency of Arg+ prototrophs generated by heteroallelic recombination. Bars show mean ± SD of six replicates; points show individual measurements. Below, Rec102 levels were assessed by anti-LexA immunoblots; anti-tubulin immunoblots served as loading controls. The shown lanes were from a single exposure of the same blot; the vertical dashed line indicates where the image was spliced to remove irrelevant lanes. The full immunoblot is provided in Source Data. **f**, Detail view of the hydrophobic interface between the amphipathic stalk and the β strands; related residues in Top6B and Rec102 are indicated.

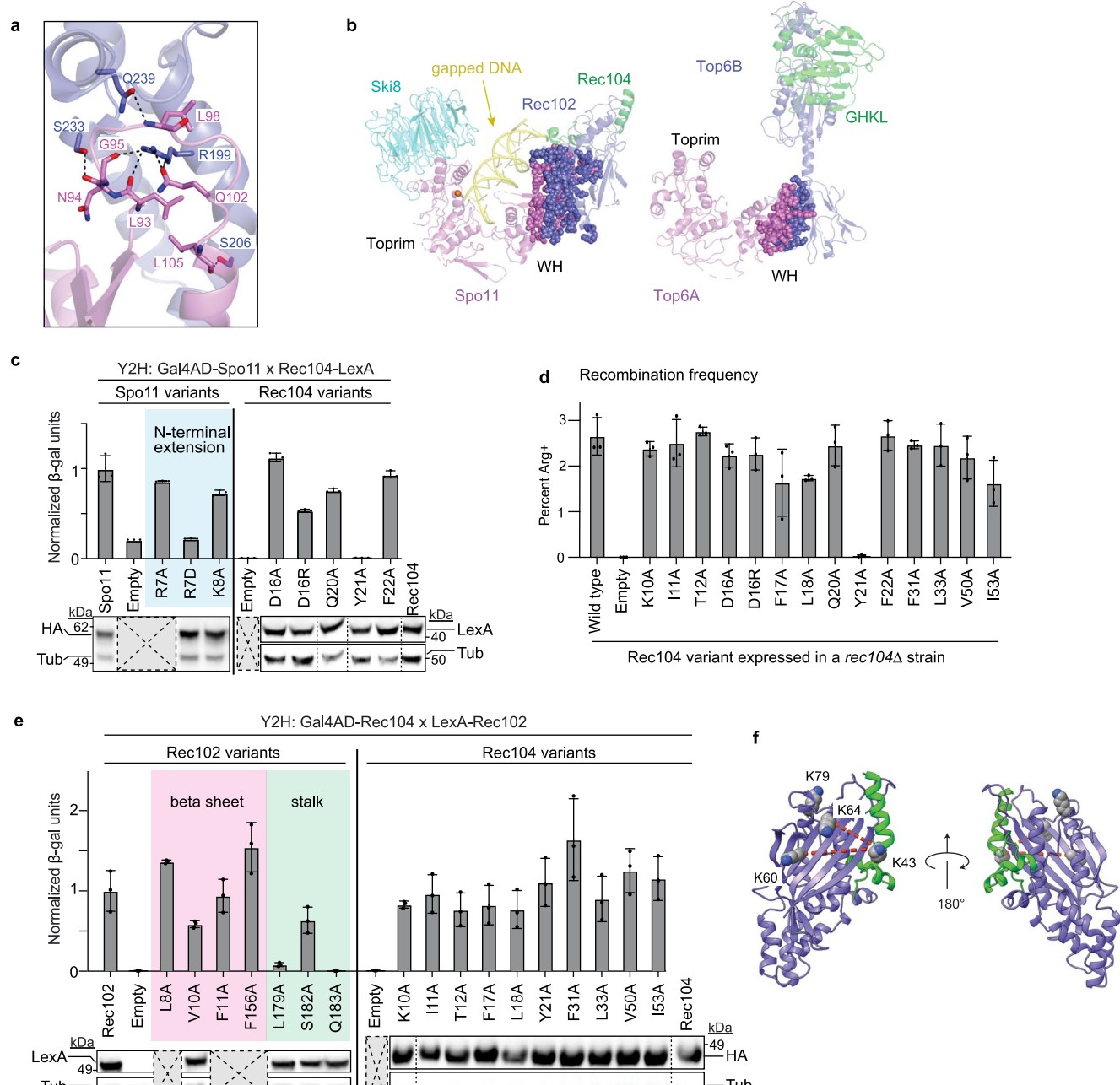

**Extended Data Fig. 8 | Functional analyses of protein-protein interfaces within the core complex. a**, Hydrogen bonding interactions between Spo11 and Rec102. **b**, Comparing interfacial areas for Spo11–Rec102 (1,680 Å², complex with gapped DNA) vs. Top6A–Top6B (*M. mazei*, 958 Å², PDB: 2q2e). Interfacial residues are shown as spheres. **c**, Y2H interactions of mutants of Gal4AD-Spo11 and Rec104-LexA. Bars show mean ± SD of three biological replicates; points show individual measurements. Immunoblots below are presented as in Fig. 3d. Several mutants yielded notable differences in recombination phenotype compared with the Y2H assay. There were two main types of such discrepancy. In one type, the mutation strongly compromised recombination without diminishing the Y2H interaction (for example, *rec102-L207A*, highlighted in yellow in Fig. 3c,d). Because the core complex presumably must do many things besides just coming together as a single multiprotein entity, it is not surprising that some mutations disrupt biological function without apparently eliminating core complex assembly. The second discrepant type is those that disrupt the Y2H interaction without disrupting recombination (for example, the *spo11-L105A*, highlighted

in peach color in Fig. 3c,d). Such mutants are more surprising than the first type since we assume that integrity of the core complex is critical for DSB formation. We do not know the explanation for these mutants, other than to consider that the artificial Y2H system may fail to recapitulate protein interactions in the more complex environment of the DSB formation machinery that can stabilize protein-protein interfaces that are weakened by some mutations. **d**, Heteroallele recombination assays with different Rec104 variants introduced into a *rec104Δ* reporter strain. Bars show mean ± SD of three biological replicates; points show individual measurements. **e**, Y2H interactions of mutants of LexA-Rec102 and Gal4AD-Rec104. Bars show mean ± SD of three biological replicates; points show individual measurements. Immunoblots below are presented as in Fig. 3d. **f**, Positions of lysines in Rec102 and Rec104 that are readily crosslinked. Red dashed lines connect the α carbons of lysines that were crosslinked in a prior study[14]. The distances between their Cα atoms are less than the 27.4 Å maximum for the crosslinker used (19.1 Å for K43 to K64; 24.3 Å for K43 to K60).

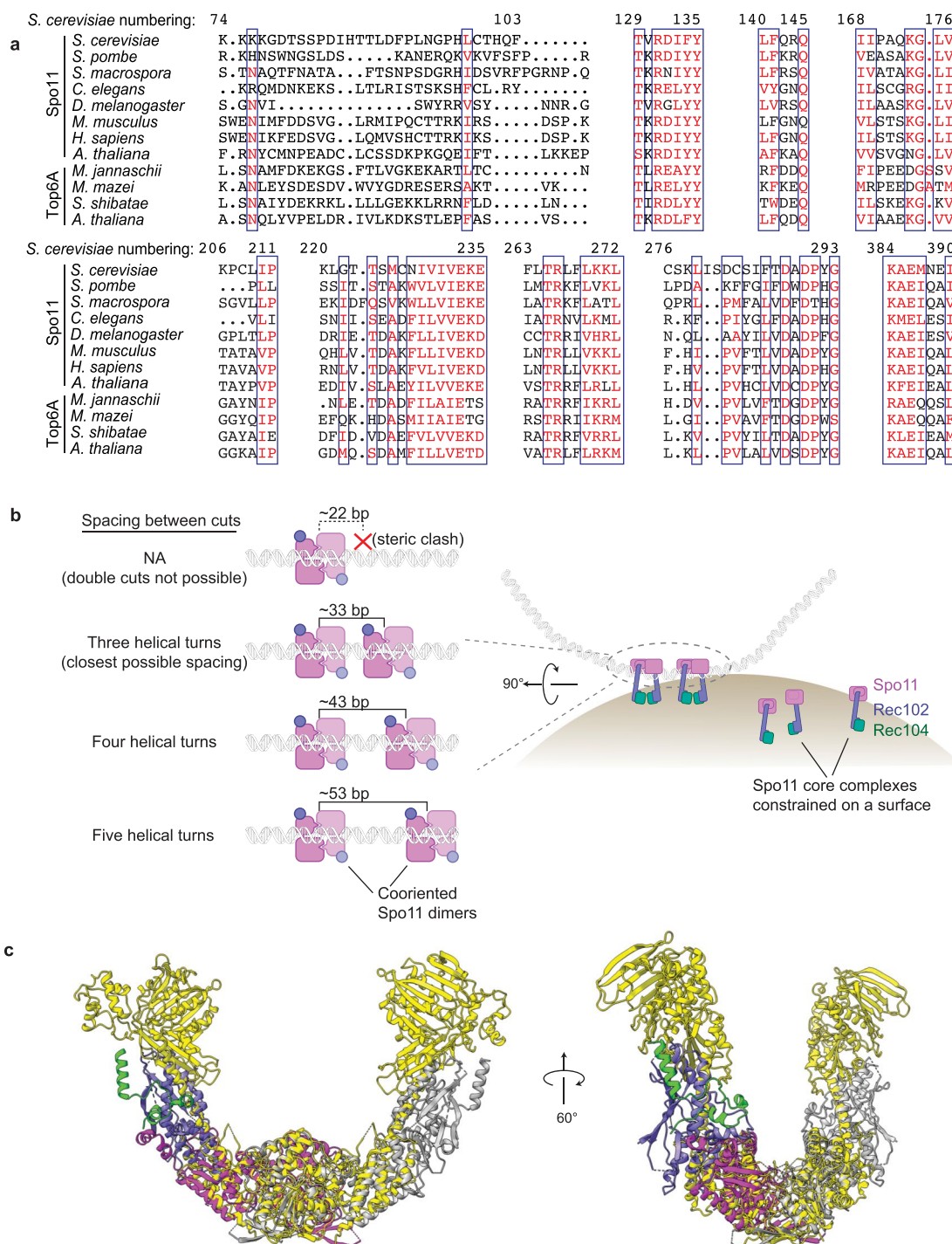

**Extended Data Fig. 9 | Spo11 sequence conservation and structural constraints on Spo11 double cuts and pre-DSB dimers. a**, Multiple sequence alignment of Spo11 and Top6A orthologs at protein-protein and protein-DNA contact sites. **b**, Cartoons illustrating how physically constraining the orientation of adjacent Spo11 core complexes can explain the minimum length and 10-bp periodicity in the size of double-cut DSB fragments[14,35]. At right, Spo11 core complexes are proposed to be associated with the surface of chromosome-associated biomolecular condensates containing Rec114, Mei4, and Mer2 proteins[14]. Dimeric core complexes can capture a DNA molecule and break it.

At left, if two adjacent dimeric complexes cut the same DNA molecule, the double cuts can be three or more helical turns apart, but not closer than that. The constrained orientation of the Spo11 core complexes further imposes a 10-bp periodicity on the allowed spacing of double cuts. **c**, Structural alignment of the hypothetical model of a Spo11 core complex pre-DSB dimer and the crystal structure of the dimeric *S. shibatae* Topo VI holoenzyme (yellow). One copy of the Spo11 core complex is colored as Fig. 1b, and the other copy is colored in gray. Ski8 is not shown for simplicity.

# Reporting Summary

## Statistics

For all statistical analyses, confirm that the following items are present in the figure legend, table legend, main text, or Methods section.

| n/a | Confirmed | |
|---|---|---|
| ☐ | ☒ | The exact sample size (*n*) for each experimental group/condition, given as a discrete number and unit of measurement |
| ☐ | ☒ | A statement on whether measurements were taken from distinct samples or whether the same sample was measured repeatedly |
| ☒ | ☐ | The statistical test(s) used AND whether they are one- or two-sided<br>*Only common tests should be described solely by name; describe more complex techniques in the Methods section.* |
| ☒ | ☐ | A description of all covariates tested |
| ☒ | ☐ | A description of any assumptions or corrections, such as tests of normality and adjustment for multiple comparisons |
| ☐ | ☒ | A full description of the statistical parameters including central tendency (e.g. means) or other basic estimates (e.g. regression coefficient) AND variation (e.g. standard deviation) or associated estimates of uncertainty (e.g. confidence intervals) |
| ☒ | ☐ | For null hypothesis testing, the test statistic (e.g. *F*, *t*, *r*) with confidence intervals, effect sizes, degrees of freedom and *P* value noted<br>*Give P values as exact values whenever suitable.* |
| ☒ | ☐ | For Bayesian analysis, information on the choice of priors and Markov chain Monte Carlo settings |
| ☒ | ☐ | For hierarchical and complex designs, identification of the appropriate level for tests and full reporting of outcomes |
| ☒ | ☐ | Estimates of effect sizes (e.g. Cohen's *d*, Pearson's *r*), indicating how they were calculated |

*Our web collection on statistics for biologists contains articles on many of the points above.*

## Software and code

Policy information about availability of computer code

| | |
|---|---|
| Data collection | Cryo-EM images were collected using SerialEM 3.9.0 (http://bio3d.colorado.edu/SerialEM/) |
| Data analysis | The following publicly available software was used: Relion v3.0.8 (https://relion.readthedocs.io/en/release-3.1/SPA_tutorial/index.html), Cryosparc v4.2.1 (https://cryosparc.com/), Phenix v1.20.1-4487 (https://phenix-online.org/), Coot 0.8.9.2 (https://www2.mrc-lmb.cam.ac.uk/personal/pemsley/coot/), PyMol v2.5.3 (https://pymol.org/2/), Chimera v1.16 (https://www.cgl.ucsf.edu/chimera/), ChimeraX v. 1.6.1 (https://www.cgl.ucsf.edu/chimerax/), GraphPad Prism v. 10.2.3 (https://www.graphpad.com/scientific-software/prism/www.graphpad.com/scientific-software/prism/), MotionCor2 (ref. 47), CTFFIND-4 (ref. 48), Colabfold (ref. 52), COBALT (ref. 45), NCBI BLAST (https://blast.ncbi.nlm.nih.gov/Blast.cgi). |

For manuscripts utilizing custom algorithms or software that are central to the research but not yet described in published literature, software must be made available to editors and reviewers. We strongly encourage code deposition in a community repository (e.g. GitHub). See the Nature Portfolio guidelines for submitting code & software for further information.

## Data

Policy information about availability of data

All manuscripts must include a data availability statement. This statement should provide the following information, where applicable:
- Accession codes, unique identifiers, or web links for publicly available datasets
- A description of any restrictions on data availability
- For clinical datasets or third party data, please ensure that the statement adheres to our policy

The atomic coordinates and cryo-EM density maps for the hairpin DNA bound to Spo11 core complex (PDB: 8URU (https://www.rcsb.org/structure/8URU); EMD: EMD-42501 (https://www.ebi.ac.uk/emdb/EMD-42501) and for gapped DNA bound to Spo11 core complex (PDB: 8URQ (https://www.rcsb.org/structure/8URQ); EMD: EMD-42497(https://www.ebi.ac.uk/emdb/EMD-42497) have been deposited in the Research Collaboratory for Structural Bioinformatics Protein Data Bank and Electron Microscopy Data Bank, respectively. We used the following published atomic coordinate accessions: PDB 2ZBK, 2Q2E,1D3Y,1S4U; AlphaFold database (https://alphafold.ebi.ac.uk/) AF-Q02721-F1-model_v1, AF-P33323-F1-model-v1, AF-P23179-F1-model_v1.

## Research involving human participants, their data, or biological material

Policy information about studies with human participants or human data. See also policy information about sex, gender (identity/presentation), and sexual orientation and race, ethnicity and racism.

| | |
|---|---|
| Reporting on sex and gender | N/A |
| Reporting on race, ethnicity, or other socially relevant groupings | N/A |
| Population characteristics | N/A |
| Recruitment | N/A |
| Ethics oversight | N/A |

Note that full information on the approval of the study protocol must also be provided in the manuscript.

# Field-specific reporting

Please select the one below that is the best fit for your research. If you are not sure, read the appropriate sections before making your selection.

☒ Life sciences ☐ Behavioural & social sciences ☐ Ecological, evolutionary & environmental sciences

For a reference copy of the document with all sections, see nature.com/documents/nr-reporting-summary-flat.pdf

# Life sciences study design

All studies must disclose on these points even when the disclosure is negative.

| | |
|---|---|
| Sample size | Cryo-EM: No statistical method was used to predetermine the sample size. The number of particles used in structural determination was not pre-determined. Sample size was arbitrarily selected based on the time required to collect data. The sample sizes are sufficient because many independently recorded images were acquired as part of cryo-EM data collection. The particle numbers were sufficient based on the multiple rounds of classification and the quality of the cryo-EM maps. For DNA binding assays, Y2H assays, and heteroallele recombination assays, sample sizes (three independent replicates) were performed in keeping with standard practice in the field for these assays. Immunoblotting was performed at least twice for each sample (this analysis was not quantitative, so additional replicates were not necessary). |
| Data exclusions | Cryo-EM: Cryo-EM images with bad ice or contamination were removed. Regarding the particle selection, 2D and 3D classification were used and criterion is based on the quality of resulting 2D class average and 3D maps. No data were excluded from DNA binding, heteroallele recombination assays and yeast two-hybrid assays. |
| Replication | Cryo-EM: At least three rounds of structural refinement have been performed and all resulted in same density maps. DNA binding assays, heteroallele recombination assays and yeast two-hybrid assays were conducted on three independent replicates with similar results. The protein purification was performed more than six times with similar results. Immunoblots were repeated at least twice with similar results. All attempts at replication were successful. |
| Randomization | Cryo-EM: Particles were randomized following extraction to avoid bias during particle classification. All other experiments involved comparison of mutants to matched wild-type controls, so randomization is neither necessary nor appropriate. |
| Blinding | No blinding was used for analysis of structural data. All other experiments involved comparison of isogenic control (wild type) and mutant yeast strains. It is not standard practice in the field to use blinding for these assays. Moreover, blinding is not necessary with this experimental design because meaningful effect sizes are larger than any likely effects of operator bias. |

# Reporting for specific materials, systems and methods

We require information from authors about some types of materials, experimental systems and methods used in many studies. Here, indicate whether each material, system or method listed is relevant to your study. If you are not sure if a list item applies to your research, read the appropriate section before selecting a response.

## Materials & experimental systems

| n/a | Involved in the study |
|-----|----------------------|
| ☐ | ☒ Antibodies |
| ☐ | ☒ Eukaryotic cell lines |
| ☒ | ☐ Palaeontology and archaeology |
| ☒ | ☐ Animals and other organisms |
| ☒ | ☐ Clinical data |
| ☒ | ☐ Dual use research of concern |
| ☒ | ☐ Plants |

## Methods

| n/a | Involved in the study |
|-----|----------------------|
| ☒ | ☐ ChIP-seq |
| ☒ | ☐ Flow cytometry |
| ☒ | ☐ MRI-based neuroimaging |

## Antibodies

| Antibodies used | Anti-Flag M2 affinity resin (Sigma Aldrich, catalog no. A2220); anti-LexA (Sigma Aldrich, catalog no. 06-719); anti-HA (Roche, catalog no. 12013819001); anti-alpha-tubulin (Invitrogen, catalog no. MA1-80017 and Santa Cruz, catalog no. sc-53030); donkey anti-rabbit IgG (Invitrogen, catalog no. A-21206); donkey anti-rat IgG (Invitrogen, catalog no. A-21209). |
|---|---|
| Validation | The flag epitope was added to the expression construct to produce recombinant Spo11 core complexes in insect cells. Specificity was confirmed by successful purification of Spo11 complexes. The HA epitope is part of the Gal4 activating domain fusion construct. Specificity of HA and LexA antibodies for immunoblotting was validated using control strains that did not express the fusion protein of interest. Tubulin antibody specificity has been established for many species in many applications (summarized at https://www.thermofisher.com/antibody/product/alpha-Tubulin-Antibody-clone-YL1-2-Monoclonal/MA1-80017 and https://www.scbt.com/p/alpha-tubulin-antibody-yol1-34) and confirmed for our experiments using immunoblotting of S. cerevisiae extracts by detection of a band of the appropriate molecular weight. |

## Eukaryotic cell lines

Policy information about cell lines and Sex and Gender in Research

| Cell line source(s) | Yeast strains were generated in the Keeney laboratory.<br>Gibco Spodoptera Frugiperda Sf9 cells for expression of recombinant proteins were from Thermo Fisher (catalog #11496015). |
|---|---|
| Authentication | Yeast strains were verified by PCR and/or Southern blotting. Mutant protein constructs were confirmed by sequencing. Sf9 description of certification analyses is available at https://www.thermofisher.com/order/catalog/product/11496015#/11496015 "Each lot of Gibco™ Sf9 cells is tested for cell growth and viability post-recovery from cryopreservation. In addition, the Master Seed Bank has been tested for contamination of bacteria, yeast, mycoplasma and virus and has been characterized by isozyme and karyotype analysis." No further authentication was performed in this study. |
| Mycoplasma contamination | Not applicable for yeast strains. Mycoplasma testing of Sf9 cells is referenced at https://www.thermofisher.com/order/catalog/product/11496015#/11496015. No further mycoplasma testing was done for this study. |
| Commonly misidentified lines<br>(See ICLAC register) | No commonly misidentified cell lines were used in this study (yeast strains and Sf9 cells only) |

## Plants

| Seed stocks | N/A |
|---|---|
| Novel plant genotypes | N/A |
| Authentication | N/A |

