## [Peer Review File · Nature Structural & Molecular Biology]

Peer Review Information

Manuscript Title: Cryo-EM structures of the Spo11 core complex bound DNA

Corresponding author name(s): Scott Keeney

Reviewer Comments & Decisions:

Decision Letter, initial version:

Message: 15th Dec 2023

Dear Dr. Keeney,

Thank you again for submitting your manuscript "Cryo-EM structure of the Spo11 core complex bound to DNA". We now have comments (below) from the 3 reviewers who evaluated your paper. In light of these reports, we remain interested in your study and would like to see your response to the comments of the referees, in the form of a revised manuscript.

You will see that all experts appreciate the novelty and importance of the findings, suggesting that this study, if appropriately revised, would constitute an important contribution to the field. However, the experts simultaneously raise important issues that need to be addressed in a revised manuscript. More specifically, reviewer #1 notes discrepancies between phenotypes of the different mutants in the functional assays used which are not adequately explained or discussed, whereas reviewer #3 requests important control experiments to validate that protein folding is not disrupted in "all of the interface mutants that show a loss of binding in Y2H experiments". Furthermore, multiple experts note that the manuscript would benefit from figure reorganisation and redesign, further discussion, additional experimental details and clarifications, providing relevant guidance on how to further highlight the novelty of the findings. Amongst these useful guidelines on further discussion and contextualisation of the novel data, which we urge you to follow, we editorially note the request by R#3 to "provide some additional insights or speculation into how Spo11 activation might be achieved".

Please be sure to address/respond to all concerns of the referees in full in a point-by-point response and highlight all changes in the revised manuscript text file. If you have comments that are intended for editors only, please include those in a separate cover letter.

We expect to see your revised manuscript within 2-3 months. If you cannot send it within this time, please contact us to discuss an extension; we would still consider your revision, provided that no similar work has been accepted for publication at NSMB or published elsewhere.

Reporting Summary:

When submitting the revised version of your manuscript, please pay close attention to our [href="https://www.nature.com/nature-portfolio/editorial-policies/image-integrity">Digital Image Integrity Guidelines](https://www.nature.com/nature-portfolio/editorial-policies/image-integrity). and to the following points below:

SOURCE DATA: we urge authors to provide, in tabular form, the data underlying the graphical representations used in figures. This is to further increase transparency in data reporting, as detailed in this editorial (<http://www.nature.com/nsmb/journal/v22/n10/full/nsmb.3110.html>). Spreadsheets can be submitted in excel format. Only one (1) file per figure is permitted; thus, for multi-paneled figures, the source data for each panel should be clearly labeled in the Excel file; alternately the data can be provided as multiple, clearly labeled sheets in an Excel file. When submitting files, the title field should indicate which figure the source data pertains to. We encourage our authors to provide source data at the revision stage, so that they

are part of the peer-review process.

Data availability: this journal strongly supports public availability of data. All data used in accepted papers should be available via a public data repository, or alternatively, as Supplementary Information. If data can only be shared on request, please explain why in your Data Availability Statement, and also in the correspondence with your editor. Please note that for some data types, deposition in a public repository is mandatory - more information on our data deposition policies and available repositories can be found below: <https://www.nature.com/nature-research/editorial-policies/reporting-standards#availability-of-data>

Nature Structural & Molecular Biology is committed to improving transparency in authorship. As part of our efforts in this direction, we are now requesting that all authors identified as 'corresponding author' on published papers create and link their Open Researcher and Contributor Identifier (ORCID) with their account on the Manuscript Tracking System (MTS), prior to acceptance. This applies to primary research papers only. ORCID helps the scientific community achieve unambiguous attribution of all scholarly contributions. You can create and link your ORCID from the home page of the MTS by clicking on 'Modify my Springer Nature account'. For more information please visit please visit www.springernature.com/orcid.

[Redacted]

Sincerely,

Dimitris Typas
Associate Editor
Nature Structural & Molecular Biology

ORCID: 0000-0002-8737-1319

Reviewers' Comments:

Reviewer #1:

Remarks to the Author:

The manuscript "Cryo-EM structure of the Spo11 core complex bound to DNA" by Yu et al. describes the long-sought after structure of Spo11 and its accessory subunits (Rec102, Rec104, and Ski8) from *S. cerevisiae*. This structure represents something of a "holy grail" to meiosis researchers. As such, I wholeheartedly support its eventual publication. The structure presented here appears to reveal a post-DNA cleavage state of the Spo11 core complex, with a "monomeric" (1:1:1:1) complex bound to a DNA end mimicking the cleaved product. How to isolate and determine a structure of the pre-cleavage "dimeric" (2:2:2:2) state of the complex is a big future goal. The current structure is, however, an important step and no doubt reveals something about how Spo11 is distinct from its archaean cousin, topoisomerase VI. As noted above, I support publication of this work after the authors address the concerns and questions noted below.

Major notes:

Overall, the structures seem well-done and well-supported. One thing the authors should note more explicitly is that, even though they observe that each DNA substrate can bind more than one Spo11 core complex in vitro (Extended Data Fig. 1b-c), they prepared the cryoEM samples with excess DNA so as to isolate only DNAs bound to single Spo11 complexes (if I read the methods correctly).

Figure 3c and Extended Data Fig. 8: I have several questions and concerns about these data and their description and interpretation. First, it would be helpful if the authors could better explain the heteroallele recombination assay in the text (in 1-2 sentences) so that a general reader will better understand what's being measured. For my own curiosity: in a strain such as Spo11 null or similar, when meiosis is expected to fail completely, how can this assay work at all? My sense is that effectively all cells would die upon sporulation, so that nothing would grow on the YPD plate after meiotic induction. Therefore, the ratio of Arg+ colonies to total colonies would be impossible to calculate since the denominator (total colonies) would be zero. What's the explanation here? Do most cells die, but some parental diploids simply not sporulate and therefore grow back after plating on YPD? Is this what "appropriate dilutions of cultures" means in the methods - that different strains had very different overall survival rates?

Next, the way the data in these figures is presented is confusing. Graphs need to be better labeled with what each assay is, what the background strain is, etc. The panel shading also makes it hard to read - these can potentially be replaced by different-colored bars to indicate the different regions? The organization of multiple graphs in a single figure panel makes things hard to find and parse. And why are there sample images for Arg+ colonies in Figure 3C? These seem disconnected from the figure and therefore confusing for a reader to understand why they are there.

Finally, and perhaps most significantly: The authors test a large set of mutations in two assays: the heteroallele recombination assay purportedly measures crossover frequency (in this case, used as a proxy for DSB formation) and the yeast two hybrid assays

measure direct protein-protein interactions. Two concerns here: first, do the Y2H assays conducted in meiotic conditions truly measure direct protein-protein interactions? What's the logic behind when the authors use meiotic versus vegetative conditions for Y2H? Secondly, a big concern is that the effects of many mutants don't agree with one another across the two assays. For example, Spo11 L105A appears to (almost) completely eliminate binding to Rec102 in Y2H, yet the recombination frequency is near-WT. Meanwhile, Rec102 L207A shows effectively no recombination, but interacts just fine with Spo11. There are many mutants like this. These inconsistencies, combined with the fact that the authors appear to mainly discuss the mutants whose phenotypes match up across the assays, plus put all the Y2H data in the supplement, make it seem like the authors are cherry-picking. Redesigning the figures to show both phenotypes for each mutant (perhaps in matched vertically-stacked graphs with heteroallele recombination on top and Y2H on bottom) in the main text would make it easier to describe and highlight the most meaningful data.

Overall, the structure description sections (especially the DNA interactions and magnesium interactions sections) are very long - non-structural readers (and even most structural experts) are likely to tune out/lose the thread here.

Lines 382-393: For a long time, I was confused by the idea of "co-orientation" of Spo11 complexes being one of the two constraints on Spo11 double cutting. It wasn't until I went back to the prior papers on double cutting and thought about it for a while, that I finally understood why a 33 base pair spacing is the smallest possible spacing (and not 22 bp). I think this needs to be made more clear in the figure describing this (Figure 6d) so that readers have a better chance of understanding by the 22 bp model - which shows no clashes - is not suitable for double cutting.

Line 454-463: On the subject of the elusive Spo11 dimer: can the authors identify anything systematically different (interface shape, surface area, hydrophobicity, etc.) about the putative Spo11 dimer interface compared to the Top6A dimer interface?

One thought that the authors likely have had (and may already be included, apologies if I missed it) is that once cleavage occurs, Spo11 may adopt a conformation that is incompatible with dimer formation. This is supported by the dimer modeling, which shows clashes if domain orientations are not adjusted (Fig. 7a). This may well be an adaptation that makes Spo11-mediated cleavage irreversible - unlike Top6A which can simply re-form the dimer and reverse DNA cleavage.

Minor notes:

Line 36-37: Note, not all Top6B proteins have the C-terminal domain, at least if the authors are referring to the C-terminal alpha-beta domain seen in *M. mazei* Top6B (<https://alphafold.ebi.ac.uk/entry/A0A0E3PZG0>). For example, *S. shibatae* top6B lacks this domain (<https://alphafold.ebi.ac.uk/entry/O05207>).

Line 216. It would be helpful to note how long Rec104 is, as in "Only residues 9-58 (of XXX) of Rec104 are visible..."

Figure 3: the overall structure figure could use its own panel label. I also recommend pulling out panels a and b from this figure and putting in supplemental. This makes room for bringing the Y2H data into main text.

Reviewer #2:

Remarks to the Author:

In this manuscript, Yu et al report cryo-EM structures of the Spo11 core complex (Spo11 - Rec102-Rec104-Ski8) bound to DNA. The Spo11 complex is responsible for forming double-strand breaks during meiosis that initiate meiotic recombination and are essential for meiotic division. Structural understanding of the Spo11 complex had been lacking, so this manuscript constitutes a landmark contribution to the field. On the basis of the solved structures, the authors make substantial conclusions regarding the basis of DNA binding and how an unusual left-handed wrapping of the core around DNA defines its footprint. They further utilise the solved structures to build a model of the pre-DSB complex, providing the first structural insight into this essential early meiotic complex. Overall, the data quality is excellent, structure solution has been performed carefully, all findings are supported by data, and figures and text are extremely clear. Owing to the excellent quality of the data and manuscript, I have no specific recommendations for improvements. I am therefore happy to recommend publication in Nature Structural & Molecular Biology in its current form.

Reviewer #3:

Remarks to the Author:

Summary

Spo11 is the conserved topoisomerase-like factor that initiates meiotic recombination by forming double-stranded DNA breaks (DSBs). While recent hybrid structural biology approaches, also from the Keeney laboratory, have provided some vital insights into the structural organisation of the Spo11-complex (Claeys Bouuaert et al, NSMB, 2021), a high-resolution experimentally determined structure has been lacking. Furthermore, the details of potential protein-DNA contacts, critical to understanding how Spo11 preferentially binds to, and breaks, certain DNA structures and sequences, could only be inferred through homology modelling. The structural work described in Yu et al is a milestone in more than 20 years of effort by Scott Keeney and colleagues. The overall organisation of Spo11-complex is confirmed, and new insights into interaction with DNA is provided. Furthermore the authors also carry out some mutational analysis, based on some novel aspects of the structure, that further boosts the impact of the manuscript. This work has the potential to be of great general interest, and has been eagerly awaited. However, in its current form the paper unfortunately falls a little flat. While technically outstanding (bar one essential control) the paper does not provide an appropriate level of insight and context, and spends too much time on relatively minor details. Thus the authors are selling their excellent data, and themselves somewhat short. Some alterations, rearrangements, plus additional analyses would improve the readability and clarity, while providing some additional context for the non-specialist reader.

Major points

For the mutations that eliminate interaction with Spo11 in Y2H (e.g. Rec102 R199A) can the authors show that this mutation doesn't completely disrupt the fold of the protein? For example, can they show that Rec102 R199A still interacts with Rec104? Ideally this control

would be done for all of the interface mutants that show a loss of binding in Y2H experiments.

I would urge the authors to reorganise the figures. There are some very important figures which are in the extended data, and space could be made by moving some panels of the current main figures to the extended data. This would create a manuscript that is more readable, and more interesting, especially for the generalist.

One of the exciting aspects of Spo11 is that, as once again shown by the authors, the recombinant Spo11 complex is not catalytically active. The authors present this as a limitation of the study, whereas in reality it is a feature. With these beautiful structures in hand the authors should provide some additional insights or speculation into how Spo11 activation might be achieved. I suggest, at the minimum, the authors examine the surface residue conservation of the complex, perform a structural comparison with AlphaFold2 predictions of Spo11 complexes from other species, and summarise known interactions between the Spo11 complex components with other parts of the meiotic DSB machinery (e.g. Rec114, Mei4 etc.).

Minor points/suggestions

Title - would "CryoEM structures...." be more appropriate, since the authors have actually determined two structures?

The authors should include, at the start of the results section, a brief rationale for their choice of DNA substrates.

Line 75 - Please provide more information on how AlphaFold2 models were generated and used. This is particularly relevant given that the authors state (line 57) the difficulty in obtaining predicted structural information on Rec104. Some information should be in the main text, details would be sufficient in the (supplementary) methods.

Lines 92 and 93 - Perhaps remind the reader that in the post-DSB state Y135 would presumably be covalently bound to DNA?

Line 115 and onwards - Validation of the Spo11-Ski8 interface. The role of Ski8 in the fungal Spo11 complexes is a curiosity. Do the cryoEM structures here provide any additional insights into what Ski8 might be doing? If the authors were to compare their structures to other Ski8 containing complexes, especially those published in the last 2 years (e.g. Keidel et al., Mol. Cell 2023) do they see anything interesting?

Lines 162-168 - Have switch loop mutants of Rec102 previously been tested for their effect on DSB formation/recombination? If not it does seem appropriate for it to be done here, unless there is a good reason not to do so.

Extended Figure 5c - Might it be helpful here to label the secondary structure elements in Rec102 as shown in Figure 3a? Especially since there are referred to collectively in lines 132-133

Figure 3a and b - It might be helpful to provide additional clarity in terms of Rec102 and Top6B and the GHKL and transducer domains, i.e. putting Figure 3a and b in a big picture context. Suggestions - copy the cartoon from Figure 1a and place this in Figure 3, labelled

appropriately. An extra supplementary figure showing a relevant structural superposition of the Spo11 cryoEM structures (or just Spo11 and Rec102) on the crystal structure of archaeal TopoVI. Actually some additional clarity would be important since the section on Rec102 and Spo11 is highly descriptive. In fact, the comparison in extended Figure 8c would be very welcome in a main figure.

Figure 6 - All of this could possibly be moved to the Extended Data. Figure 6d is a little confusing at first, especially to those readers who are not familiar with the whole "double-cut" data. The models superficially look like those shown in Figure 7e, i.e. the active Spo11 dimer making DSBs. If 6d is kept as a main figure please somehow make it clear how it is different from 7e.

Extended Data Figure 8a - Part of this data belongs in the main text, at least as some kind of summary for the outcome of the interaction mutational Y2H analysis.

It might be interesting to compare, in an extra figure, the experimental chemical cross-links previously determined by the authors on a Spo11-complex, to the experimental structures here. The authors do refer to these cross-linking data in the main text, but an extra figure would be helpful.

Author Rebuttal to Initial comments

We thank the reviewers for the constructive and positive feedback. We have addressed nearly all of the critiques with revised text, reorganization of the figures and text, or addition of new data. Black text below is the reviewer comments; our responses are in blue. Line numbers below refer to the version of the manuscript without changes tracked.

Reviewers' Comments:

Reviewer #1:

Remarks to the Author:

The manuscript "Cryo-EM structure of the Spo11 core complex bound to DNA" by Yu et al. describes the long-sought after structure of Spo11 and its accessory subunits (Rec102, Rec104, and Ski8) from *S. cerevisiae*. This structure represents something of a "holy grail" to meiosis researchers. As such, I wholeheartedly support its eventual publication. The structure presented here appears to reveal a post-DNA cleavage state of the Spo11 core complex, with a "monomeric" (1:1:1:1) complex bound to a DNA end mimicking the cleaved product. How to isolate and determine a structure of the pre-cleavage "dimeric" (2:2:2:2) state of the complex is a big future goal. The current structure is, however, an important step and no doubt reveals something about how Spo11 is distinct from its archaean cousin, topoisomerase VI. As noted above, I support publication of this work after the authors address the concerns and questions noted below.

We appreciate the positive assessment as well as the detailed feedback below.

Major notes:

Overall, the structures seem well-done and well-supported. One thing the authors should note more explicitly is that, even though they observe that each DNA substrate can bind more than one Spo11 core complex in vitro (Extended Data Fig. 1b-c), they prepared the cryoEM samples with excess DNA so as to isolate only DNAs bound to single Spo11 complexes (if I read the methods correctly).

The reviewer is correct that DNA was in excess, but it was only by twofold. While this would favor singly-bound complexes, as the reviewer notes, this should still allow recovery of doubly-bound DNA molecules if the affinity for binding by the second core complex were comparable to binding by the first. For the hairpin substrate, we already knew that the substrate is too small to favor two core complexes binding, and that binding to the hairpin end is much lower affinity, so only singly bound complexes were expected. For the gapped substrate, however, we did not know whether a pair of core complexes might be able to bind. We added text to explain in Methods (lines 496–502).

Figure 3c and Extended Data Fig. 8: I have several questions and concerns about these data and their description and interpretation. First, it would be helpful if the authors could better explain the heteroallele recombination assay in the text (in 1-2 sentences) so that a general reader will better understand what's being measured. For my own curiosity: in a strain such as Spo11 null or similar, when meiosis is expected to fail completely, how can this assay work at all? My sense is that effectively all cells would die upon sporulation, so that nothing would grow on the YPD plate after meiotic induction. Therefore, the ratio of Arg⁺ colonies to total colonies would be impossible to calculate since the denominator (total colonies) would be zero. What's the explanation here? Do most cells die, but some parental diploids simply not sporulate and therefore grow back after plating on YPD? Is this what "appropriate dilutions of cultures" means

in the methods - that different strains had very different overall survival rates?

Sorry for the lack of clarity. We added a sentence to the main text to explain what the recombination assay is (lines 168–170) and provided a schematic in the extended data (ED Fig 7d). This widely used assay is often referred to as “return to growth” because yeast cells transferred from sporulation medium to rich medium during meiotic prophase will return to vegetative (mitotic) growth without doing the meiotic divisions. This is why *spo11* null and similar DSB-defective mutants are viable: the cells are returned to rich medium before the lethal attempt to carry out the first meiotic division. The term “appropriate dilutions” just alludes to the fact that different strains require different dilutions to give Arg⁺ colony numbers appropriate for counting, and to the need for different dilutions plated on rich medium (where nearly all cells make a colony) vs. medium lacking arginine (where only a small percentage of cells grows).

Next, the way the data in these figures is presented is confusing. Graphs need to be better labeled with what each assay is, what the background strain is, etc. The panel shading also makes it hard to read - these can potentially be replaced by different-colored bars to indicate the different regions? The organization of multiple graphs in a single figure panel makes things hard to find and parse. And why are there sample images for Arg⁺ colonies in Figure 3C? These seem disconnected from the figure and therefore confusing for a reader to understand why they are there.

Thanks for the feedback. To make these graphs more understandable, we added better explanatory labels, we split the graphs up into separate panels, we lightened the background shading to enhance contrast and make the graphs more readable, and we moved the images of plates to the supplement to accompany the schematic of the recombination assay (new ED Fig. 7d).

Finally, and perhaps most significantly: The authors test a large set of mutations in two assays: the heteroallele recombination assay purportedly measures crossover frequency (in this case, used as a proxy for DSB formation) and the yeast two hybrid assays measure direct protein-protein interactions. Two concerns here: first, do the Y2H assays conducted in meiotic conditions truly measure direct protein-protein interactions? What’s the logic behind when the authors use meiotic versus vegetative conditions for Y2H?

Thanks for these points; we agree that we could have presented these more clearly. Y2H assays do not necessarily reflect just those interactions that are direct, and we did not mean to imply that they do. It is simply one (imperfect) way to measure the impact of the mutations on protein-protein interactions, whether direct or supported by interactions with other proteins present in the reporter cells. We pioneered the use of meiotic reporter strains 20 years ago to allow us to detect protein-protein interactions that only occur in the context of meiosis (Arora et al. Mol Cell 2004). For the interaction pairs that require a meiotic Y2H assay analyzed here (Spo11 with either Rec102 or Rec104), we previously showed that presence of other members of the core complex is needed (e.g., interaction of Spo11 with Rec102 requires that Rec104 also be present) (Maleki et al. Chromosoma 2007). We added information to the legend of ED Fig. 8a to explain this.

Secondly, a big concern is that the effects of many mutants don’t agree with one another across the two assays. For example, Spo11 L105A appears to (almost) completely eliminate binding to Rec102 in Y2H, yet the recombination frequency is near-WT. Meanwhile, Rec102 L207A shows effectively no recombination, but interacts just fine with Spo11. There are many mutants like this. These inconsistencies, combined with the fact that the authors appear to mainly discuss

the mutants whose phenotypes match up across the assays, plus put all the Y2H data in the supplement, make it seem like the authors are cherrypicking.

Thanks for highlighting the insufficient clarity. Of course, our intention here was the opposite of cherry picking: We're presenting all of the data, even the uninformative results. The mutations that didn't have an effect don't really tell us anything, so there was nothing for us to comment on. We included them for full transparency.

We originally chose not to elaborate on the cases where the Y2H and recombination results differ because many of these require more investigation to understand. However, we recognize that not saying anything can be confusing, so we updated the manuscript (particularly the legend to ED Fig. 8). There are two types of discrepancy between recombination frequencies and Y2H. One type is fairly easy to understand in principle: mutants that kill recombination without killing the Y2H interaction (e.g., the *rec102-L207A* mutation highlighted by the reviewer). We know that the core complex has to do a lot of things besides just coming together as a single multiprotein entity, so it is not surprising that we have uncovered some mutations that disrupt biological function without apparently eliminating core complex assembly. These are potentially interesting mutants, but we don't have enough information to know what they are telling us yet.

The second discrepant type is those that disrupt the Y2H interaction without killing recombination (e.g., the *spo11-L105A* mutant highlighted by the reviewer). Such mutants are more surprising than the first type since we assume that integrity of the core complex is critical for DSB formation. In some cases, (e.g., *rec102-Q183A*, which completely disrupts the Rec104 Y2H interaction but has no effect on recombination), it is likely that the discrepancy arises from the Y2H assay being done in vegetative cells that lack expression of Spo11 and other components of the meiotic DSB machinery. A plausible scenario is that this mutation weakens the Rec102–Rec104 interface, but that this interaction defect can be functionally suppressed by presence of Spo11 or other meiosis-specific proteins. We suspect that other discrepancies between the assays trace to related explanations, i.e., that the artificial Y2H system sometimes fails to recapitulate protein interactions that occur in the more complex environment of the DSB formation machinery and that can stabilize protein-protein interfaces that are weakened by some mutations. We added text in the relevant sections to explain these points (e.g., lines 250–255).

Redesigning the figures to show both phenotypes for each mutant (perhaps in matched vertically-stacked graphs with heteroallele recombination on top and Y2H on bottom) in the main text would make it easier to describe and highlight the most meaningful data.

We prefer not to stack the recombination and Y2H graphs as suggested (and as we did with a much smaller and simpler set of mutants in a previous paper (Claeys Bouuaert et al. NSMB 2021)) because we think that the recombination data are the most important (because they are the most physiologically relevant) and we prefer to keep that data organized in parallel to the flow of the text, which is by protein and by protein subdomain. When organized this way, the Y2H data can't be plotted in strictly parallel order because different mutations within the same subdomain were tested against different Y2H partners. Recognizing that no solution is perfect, we think it makes the most sense to present the recombination and Y2H data in separate graphs. However, we have reorganized the figures to move Y2H data into the main figure, and the shading is matched between the figures to provide landmarks to facilitate comparisons.

Overall, the structure description sections (especially the DNA interactions and magnesium

interactions sections) are very long - non-structural readers (and even most structural experts) are likely to tune out/lose the thread here.

We made edits to streamline the indicated sections. However, we note that this is the first report of the structure of any member of this family (either Spo11 or Topo VI) bound to DNA, and the first structure of any kind for any Spo11 member, so we think it is important to provide a comprehensive description of the results, even if it means that some readers will be interested in only limited subsets of the findings.

Lines 382-393: For a long time, I was confused by the idea of “co-orientation” of Spo11 complexes being one of the two constraints on Spo11 double cutting. It wasn't until I went back to the prior papers on double cutting and thought about it for a while, that I finally understood why a 33 base pair spacing is the smallest possible spacing (and not 22 bp). I think this needs to be made more clear in the figure describing this (Figure 6d) so that readers have a better chance of understanding by the 22 bp model - which shows no clashes - is not suitable for double cutting.

Thanks for the comment. We clarified the main text (lines 358–365 and added an explanatory cartoon (new Extended Data Fig. 9b). We were trying to avoid an extensive recap of the published discussion of this phenomenon, but we agree that we needed to provide more explanation than we did originally. Hopefully, this is now clearer.

Line 454-463: On the subject of the elusive Spo11 dimer: can the authors identify anything systematically different (interface shape, surface area, hydrophobicity, etc.) about the putative Spo11 dimer interface compared to the Top6A dimer interface?

So far, the answer is unfortunately no. We have compared the dimer interfaces for Spo11 family members with Top6A proteins for surface hydrophobicity, conservation of (predicted or experimentally validated) surface residues, and electrostatics, but no features emerged that clearly distinguish Spo11 from Top6A. From our analyses so far, Top6A family members seem to differ from one another about as much as Spo11 differs from Top6A.

One thought that the authors likely have had (and may already be included, apologies if I missed it) is that once cleavage occurs, Spo11 may adopt a conformation that is incompatible with dimer formation. This is supported by the dimer modeling, which shows clashes if domain orientations are not adjusted (Fig. 7a). This may well be an adaptation that makes Spo11-mediated cleavage irreversible - unlike Top6A which can simply re-form the dimer and reverse DNA cleavage.

We have indeed been thinking along these lines, so thank you for the comment. We added mention of this idea (lines 414–416).

Minor notes:

Line 36-37: Note, not all Top6B proteins have the C-terminal domain, at least if the authors are referring to the C-terminal alpha-beta domain seen in *M. mazei* Top6B (<https://alphafold.ebi.ac.uk/entry/A0A0E3PZG0>). For example, *S. shibatae* top6B lacks this domain (<https://alphafold.ebi.ac.uk/entry/O05207>).

Thanks for the correction. We amended the text to indicate that the CTD is not universal (line 33).

Line 216. It would be helpful to note how long Rec104 is, as in “Only residues 9-58 (of XXX) of Rec104 are visible...”

Done (line 230)

Figure 3: the overall structure figure could use its own panel label. I also recommend pulling out panels an and b from this figure and putting in supplemental. This makes room for bringing the Y2H data into main text.

We gave the overall structure figure its own panel label as suggested. We have reorganized the presentation of the biological data, including moving the most important Y2H data into the main figure, but we prefer to keep panels a and b in the main figure.

Reviewer #2:

Remarks to the Author:

In this manuscript, Yu et al report cryo-EM structures of the Spo11 core complex (Spo11 - Rec102-Rec104-Ski8) bound to DNA. The Spo11 complex is responsible for forming double-strand breaks during meiosis that initiate meiotic recombination and are essential for meiotic division. Structural understanding of the Spo11 complex had been lacking, so this manuscript constitutes a landmark contribution to the field. On the basis of the solved structures, the authors make substantial conclusions regarding the basis of DNA binding and how an unusual left-handed wrapping of the core around DNA defines its footprint. They further utilise the solved structures to build a model of the pre-DSB complex, providing the first structural insight into this essential early meiotic complex. Overall, the data quality is excellent, structure solution has been performed carefully, all findings are supported by data, and figures and text are extremely clear. Owing to the excellent quality of the data and manuscript, I have no specific recommendations for improvements. I am therefore happy to recommend publication in Nature Structural & Molecular Biology in its current form.

Thank you for the positive assessment

Reviewer #3:

Remarks to the Author:

Summary

Spo11 is the conserved topoisomerase-like factor that initiates meiotic recombination by forming double-stranded DNA breaks (DSBs). While recent hybrid structural biology approaches, also from the Keeney laboratory, have provided some vital insights into the structural organisation of the Spo11-complex (Claeys Bouuaert et al, NSMB, 2021), a high-resolution experimentally determined structure has been lacking. Furthermore, the details of potential protein-DNA contacts, critical to understanding how Spo11 preferentially binds to, and breaks, certain DNA structures and sequences, could only be inferred through homology modelling. The structural work described in Yu et al is a milestone in more than 20 years of effort by Scott Keeney and colleagues. The overall organisation of Spo11-complex is confirmed, and new insights into interaction with DNA is provided. Furthermore the authors also carry out some mutational analysis, based on some novel aspects of the structure, that further boosts the impact of the

manuscript. This work has the potential to be of great general interest, and has been eagerly awaited. However, in its current form the paper unfortunately falls a little flat. While technically outstanding (bar one essential control) the paper does not provide an appropriate level of insight and context, and spends too much time on relatively minor details. Thus the authors are selling their excellent data, and themselves somewhat short. Some alterations, rearrangements, plus additional analyses would improve the readability and clarity, while providing some additional context for the non-specialist reader.

Response: We appreciate the constructive feedback. We have reorganized and updated the manuscript and added controls to address whether mutations grossly destabilized the proteins, as requested.

Major points

For the mutations that eliminate interaction with Spo11 in Y2H (e.g. Rec102 R199A) can the authors show that this mutation doesn't completely disrupt the fold of the protein? For example, can they show that Rec102 R199A still interacts with Rec104? Ideally this control would be done for all of the interface mutants that show a loss of binding in Y2H experiments.

We addressed this by performing immunoblots for nearly all of the mutants for all three proteins, including all of the ones that behaved like nulls. (We omitted a few that behaved like wild type in either the recombination or Y2H assays). None of the Rec102 and Rec104 mutants tested showed reduced protein levels, indicating that these did not substantially disrupt protein folding. Most of the Spo11 mutants likewise gave similar protein levels as wild type. However, three Spo11 mutants (L6A, L53A, and L98A) gave considerably reduced protein levels and a fourth (R13A) gave irreproducible behavior among multiple replicate blots and across multiple independent clones. We removed these four mutants from the paper because they are not informative about the interaction surfaces.

I would urge the authors to reorganise the figures. There are some very important figures which are in the extended data, and space could be made by moving some panels of the current main figures to the extended data. This would create a manuscript that is more readable, and more interesting, especially for the generalist.

We followed the specific suggestions as indicated below, including moving the most important Y2H data into the main figure (now Fig. 3d) along with the new immunoblotting results, and reorganizing ED Fig. 8.

One of the exciting aspects of Spo11 is that, as once again shown by the authors, the recombinant Spo11 complex is not catalytically active. The authors present this as a limitation of the study, whereas in reality it is a feature. With these beautiful structures in hand the authors should provide some additional insights or speculation into how Spo11 activation might be achieved. I suggest, at the minimum, the authors examine the surface residue conservation of the complex, perform a structural comparison with AlphaFold2 predictions of Spo11 complexes from other species, and summarise known interactions between the Spo11 complex components with other parts of the meiotic DSB machinery (e.g. Rec114, Mei4 etc.).

We share the reviewers enthusiasm for these aspects. We added additional speculation at the end as suggested, and highlighted potential contributions of Rec114, Mei4, etc. (final paragraph). We have already provided an extensive evolutionary comparison of Spo11 core complexes from various species and of Spo11 core complexes with Topo VI, including

comparisons of AlphaFold2 predictions (Fig. 3a,b; Fig. 4d,e; Fig. 6b; ED Fig. 7; and ED Fig. 9). In response to a similar suggestion from reviewer 1 about surface conservation, we analyzed the surface conservation, charge density, and hydrophobicity of the core complex dimer interface. Unfortunately, however, these analyses have not yet illuminated this particular question, because so far it appears that Top6A family members differ from one another in these respects about as much as Spo11 differs from Top6A. We elected not to include these uninformative results.

Minor points/suggestions

Title - would “CryoEM structures...” be more appropriate, since the authors have actually determined two structures?

Changed as suggested.

The authors should include, at the start of the results section, a brief rationale for their choice of DNA substrates.

Agreed, changed as suggested (lines 65–68 and 99–102).

Line 75 - Please provide more information on how AlphaFold2 models were generated and used. This is particularly relevant given that the authors state (line 57) the difficulty in obtaining predicted structural information on Rec104. Some information should be in the main text, details would be sufficient in the (supplementary) methods.

We added more details about the models to the Methods (lines 531–534).

Lines 92 and 93 - Perhaps remind the reader that in the post-DSB state Y135 would presumably be covalently bound to DNA?

Done (lines 92–93).

Line 115 and onwards - Validation of the Spo11-Ski8 interface. The role of Ski8 in the fungal Spo11 complexes is a curiosity. Do the cryoEM structures here provide any additional insights into what Ski8 might be doing? If the authors were to compare their structures to other Ski8 containing complexes, especially those published in the last 2 years (e.g. Keidel et al., Mol. Cell 2023) do they see anything interesting?

We agree that the curiosity of the (apparently) fungus-specific role of Ski8 is interesting. Unfortunately, however, no additional insights have emerged so far. The newer structures of Ski8-containing complexes cited by the reviewer are very similar in the relevant aspects to the ones that were already available, so those comparisons have not told us anything new.

Lines 162-168 - Have switch loop mutants of Rec102 previously been tested for their effect on DSB formation/recombination? If not it does seem appropriate for it to be done here, unless there is a good reason not to do so.

The switch loop in topoisomerases is intimately tied to the GHKL domain's ATPase activity. We did not consider it a high priority to evaluate the function of the Rec102 switch loop since there is no obvious GHKL domain and the switch loop's amino acid sequence, length and AlphaFold-predicted conformation are all so variable (ED Fig. 7a,b). Nonetheless, we performed the

requested experiment. Neither of two double- or triple-alanine substitutions tested had a detectable effect on recombination *in vivo*, and even a complete replacement of the entire switch loop with a gly/ser linker retained about 25% of normal activity (new ED Fig. 7e).

Extended Figure 5c - Might it be helpful here to label the secondary structure elements in Rec102 as shown in Figure 3a? Especially since there are referred to collectively in lines 132-133

Done.

Figure 3a and b - It might be helpful to provide additional clarity in terms of Rec102 and Top6B and the GHKL and transducer domains, i.e. putting Figure 3a and b in a big picture context. Suggestions - copy the cartoon from Figure 1a and place this in Figure 3, labelled appropriately. An extra supplementary figure showing a relevant structural superposition of the Spo11 cryoEM structures (or just Spo11 and Rec102) on the crystal structure of archaeal TopoVI. Actually some additional clarity would be important since the section on Rec102 and Spo11 is highly descriptive. In fact, the comparison in extended Figure 8c would be very welcome in a main figure.

As suggested, we copied the domain organization cartoon from Fig. 1 (new Fig. 3a). A direct superposition of core complexes with Topo VI is messy because Rec102 is so contorted relative to Top6B and because the WH and Toprim domains of Spo11 are in different relative configuration than in Top6A. We develop these comparisons in considerable detail later in the manuscript, so at this point in the paper we thought it would work better to put the cartoons side by side with ribbon tracings (similar to the original ED Fig. 8c) of the two “monomeric” protein complexes (new Fig. 3a).

Figure 6 - All of this could possibly be moved to the Extended Data. Figure 6d is a little confusing at first, especially to those readers who are not familiar with the whole “double-cut” data. The models superficially look like those shown in Figure 7e, i.e. the active Spo11 dimer making DSBs. If 6d is kept as a main figure please somehow make it clear how it is different from 7e.

Thanks for the feedback. We prefer to keep this analysis as a main figure because we feel that our insights into double cutting—and what it reveals about DSB formation *in vivo*—are an important advance. We added a cartoon to ED Fig. 9 to explain the model in which spatially constraining adjacent Spo11 complexes imposes a periodicity and minimum spacing on double-cut fragment lengths. We also added a cartoon to Fig. 6d to make it clear that we are modeling how two Spo11 core complexes can fit back-to-back with one another, distinct from the model of a head-to-head pre-DSB dimer in Fig. 7. We hope that these illustrative elements make the presentation more accessible.

Extended Data Figure 8a - Part of this data belongs in the main text, at least as some kind of summary for the outcome of the interaction mutational Y2H analysis.

As suggested, we moved most of the Y2H data into the main figure (Fig. 3d) and provided more thorough summary of specific findings in the text.

It might be interesting to compare, in an extra figure, the experimental chemical cross-links previously determined by the authors on a Spo11-complex, to the experimental structures here.

The authors do refer to these cross-linking data in the main text, but an extra figure would be helpful.

We previously provided detailed information about the crosslinks between Rec102 and Rec104. We felt these were the most important because they speak to the position of Rec104 relative to the place that a GHKL domain would be expected. Since we already covered all of the crosslinks relevant to this interaction, we did not make further additions.

In our prior publication (Claeys Bouuaert et al. NSMB 2021), we provided an extensive analysis of the intramolecular crosslinks in Spo11 and Ski8 and the intermolecular crosslinks between them. Because our new structure agrees well with our prior model for the details relevant to these points, the interpretations of the crosslinking results have not materially changed. We elected not to repeat that analysis here.

Unfortunately, the remaining crosslinking data give us little else to add to the paper. There were relatively few crosslinks involving Rec102 and/or Rec104 (compared to Spo11 and Ski8), and most of the ones in Rec104 involved lysines for which we don't know the structure. There were two interpretable intramolecular crosslinks in Rec102 and three in Rec104. These crosslinks agree well with the structure, so we added mention of these to the text (lines 138–139 for Rec102; lines 230–232 for Rec104). We did not feel that these small points merited a figure. Finally, the crosslinks of Spo11 with Rec102 or Rec104 are currently uninterpretable because they involve residues that are missing from our structure.

Decision Letter, first revision:

Message: Our ref: NSMB-A48466A

22nd May 2024

Dear Dr. Keeney,

Thank you for submitting your revised manuscript "Cryo-EM structures of the Spo11 core complex bound to DNA" (NSMB-A48466A). It has now been seen by the original referees and their comments are below. The reviewers find that the paper has improved in revision, and therefore we are happy to accept it in principle in Nature Structural & Molecular Biology, pending minor revisions to satisfy the referees' final requests and to comply with our editorial and formatting guidelines.

We are now performing detailed checks on your paper and will send you a checklist detailing our editorial and formatting requirements in about two weeks. Please do not upload the final materials and make any revisions until you receive this additional information from us.

To facilitate our work at this stage, it is important that we have a copy of the main text as a word file. If you could please send along a word version of this file as soon as possible, we would greatly appreciate it; please make sure to copy the NSMB account (cc'ed above).

Sincerely,

Dimitris Typas
Senior Editor
Nature Structural & Molecular Biology
ORCID: 0000-0002-8737-1319

Reviewer #1 (Remarks to the Author):

The authors have addressed all my concerns. Congratulations on a very nice (and long-awaited!) study.

Reviewer #3 (Remarks to the Author):

The authors have addressed all of my previous concerns and questions. I think the manuscript has been strengthened, and is much more exciting for a general audience. I fully support publication in its current form.

Final Decision Letter:

Message: 1st Aug 2024

Dear Dr. Keeney,

We are now happy to accept your revised paper "Cryo-EM structures of the Spo11 core complex bound to DNA" for publication as an Article in Nature Structural & Molecular Biology.

Your paper will be published online soon after we receive proof corrections and will appear

in print in the next available issue. You can find out your date of online publication by contacting the production team shortly after sending your proof corrections.

Please note that *Nature Structural & Molecular Biology* is a Transformative Journal (TJ). Authors may publish their research with us through the traditional subscription access route or make their paper immediately open access through payment of an article-processing charge (APC). Authors will not be required to make a final decision about access to their article until it has been accepted. Find out more about Transformative Journals

Authors may need to take specific actions to achieve compliance with funder and institutional open access mandates. If your research is supported by a funder that requires immediate open access (e.g. according to Plan S principles) then you should select the gold OA route, and we will direct you to the compliant route where possible. For authors selecting the subscription publication route, the journal's standard licensing terms will need to be accepted, including self-archiving policies. Those licensing terms will supersede any other terms that the author or any third party may assert apply to any

version of the manuscript.

Sincerely,

Dimitris Typas
Senior Editor
Nature Structural & Molecular Biology
ORCID: 0000-0002-8737-1319